# A common neuronal ensemble in nucleus accumbens regulates pain-like behaviour and sleep

Haiyan Sun [1,2,4], Zhilin Li[1,4], Zhentong Qiu[1], Yu Shen[1], Qingchen Guo[1], Su-Wan Hu[1], Hai-Lei Ding[1], Shuming An [1,5] ✉ & Jun-Li Cao [1,3,5] ✉

A comorbidity of chronic pain is sleep disturbance. Here, we identify a dual-functional ensemble that regulates both pain-like behaviour induced by chronic constrictive injury or complete Freund's adjuvant, and sleep wakefulness, in the nucleus accumbens (NAc) in mice. Specifically, a select population of NAc neurons exhibits increased activity either upon nociceptive stimulation or during wakefulness. Experimental activation of the ensemble neurons exacerbates pain-like (nociceptive) responses and reduces NREM sleep, while inactivation of these neurons produces the opposite effects. Furthermore, NAc ensemble primarily consists of D1 neurons and projects divergently to the ventral tegmental area (VTA) and preoptic area (POA). Silencing an ensemble innervating VTA neurons selectively increases nociceptive responses without affecting sleep, whereas inhibiting ensemble-innervating POA neurons decreases NREM sleep without affecting nociception. These results suggest a common NAc ensemble that encodes chronic pain and controls sleep, and achieves the modality specificity through its divergent downstream circuit targets.

Clinically, up to 88% of chronic pain patients suffer from sleep disturbance[1–4], and ~50% of individuals with insomnia report pain symptoms[5–11]. It is thought that, through this comorbidity, chronic pain and sleep disturbance conditions aggravate each other, resulting in progressively deteriorating symptoms[11–16]. However, the neural basis substantiating this comorbidity remains poorly understood, preventing the development of effective clinical intervention.

It has long been hypothesized that central neurons are organized into functionally distinct ensembles or engrams, such that a select population of neurons encodes a specific behavioral output[17–20]. While this hypothesis emphasizes the functional distinction between neuronal ensembles, increasing evidence suggests that some ensembles are multi-functional[21,22], raising the possibility that a common neuronal ensemble simultaneously encodes both nociception and sleep, which belong to two distinct physiological modalities. We explored this possibility by focusing on the nucleus accumbens (NAc). The NAc receives both sensorimotor- and sleep-related inputs and projects to several brain regions that process nociception and sleep[23–25]. In addition to motivated responding, principal NAc neurons in vivo exhibit patterned activities during either chronic pain conditions or sleep-wakefulness cycles[15,26–28]. Conversely, experimental manipulation of NAc neurons changes both nociceptive responses and NREM sleep[26,29]. These previous results prompted us to search for a shared NAc ensemble that regulates both pain-like behaviors and sleep.

Using tTA-TRE and c-Fos-based dual-labeling approaches[30], we detected a dual-encoding ensemble, a select population of NAc

[1]Jiangsu Province Key Laboratory of Anesthesiology & Jiangsu Province Key Laboratory of Anesthesia and Analgesia Application Technology, NMPA Key Laboratory for Research and Evaluation of Narcotic and Psychotropic Drugs, Xuzhou Medical University, Xuzhou 221004, China. [2]Department of Pediatrics, The Affiliated Hospital of Xuzhou Medical University, Xuzhou 221006 Jiangsu, China. [3]Department of Anesthesiology, The Affiliated Hospital of Xuzhou Medical University, Xuzhou 221006, China. [4]These authors contributed equally: Haiyan Sun, Zhilin Li. [5]These authors jointly supervised this work: Shuming An, Jun-Li Cao. ✉e-mail: shumingan@foxmail.com; caojl0310@aliyun.com

neurons, that exhibited increased activities during either pain-eliciting periods or wakefulness. Selective activation of the ensemble neurons exacerbated pain-like responses and reduced NREM sleep, while inactivation of these neurons produced the opposite effects. We further identified that neurons innervated by the ensemble neurons in the ventral tegmental area regulated pain-like responses without affecting sleep, whereas ensemble-innervating neurons in the preoptic area regulated sleep without affecting pain-like responses. These results demonstrate a common NAc ensemble that encodes both pain-like behaviors and sleep information and achieves the modality specificity of pain versus sleep through its divergent downstream projections.

## Results

### The dual-functioning NAc ensemble

Both preclinical and clinical studies detect that chronic pain is often accompanied by sleep loss[1,7,8,10,12,13]. We modeled this in mice with chronic constrictive injury (CCI) (Supplementary Fig. 1a). Specifically, CCI mice exhibited persistent (up to 21-day) decreases in paw withdrawal latencies (PWL) and paw withdrawal thresholds (PWT) compared to sham mice (Supplementary Fig. 1b, c). In mice with 3-week of CCI (Supplementary Fig. 1a), electroencephalographic (EEG)/electromyographic (EMG) recordings revealed decreased NREM sleep and increased wakefulness compared to sham mice (Supplementary Fig. 1d–g). Thus, pain-like symptoms and sleep disturbance co-existed in CCI mice.

The c-Fos has been used as a neural marker of pain over several decades since Hunt et al.[31] CCI mice exhibited persistent pain hypersensitivity at least for 3 weeks. The c-Fos expression level in NAc three weeks after CCI surgery was dramatically higher than that after sham surgery (Supplementary Fig. 2a, b). Recently, the tTA-TRE (Tet-Off)-based viral system has been applied to label itch- and pain-specific neurons[30]. We also used the Tet-Off system in c-Fos-tTA mice[32,33] to identify and manipulate pain- and sleep-associated NAc neurons in present study (Fig. 1a). We unilaterally injected these mice with adeno-associated virus 2 (AAV2-Tre-tight-ChR2-mCherry) into NAc and terminated the doxycycline supply between day 0 and 21 after CCI surgery, such that mCherry-tagged Channel Rhodopsin 2 (ChR2) were preferentially expressed in NAc neurons with high expression of c-Fos, a molecular proxy of pain-induced neuronal activities (Fig. 1a). In double-labeling fluorescence in situ hybridization (FISH) after 21 days of CCI, we observed that the mCherry signals and, thus, ChR2 were exclusively expressed in c-Fos+ NAc neurons (96.79 ± 1.05% of ChR2-mCherry neurons were c-Fos+ and 86.09 ± 1.41% of c-Fos+ neurons expressed ChR2-mCherry; Fig. 1b). Thus, ChR2 was preferentially expressed in CCI-activated NAc neurons through this approach.

We next performed optogenetic tagging and optrode recordings[34] to monitor ChR2-expressing NAc neurons during the sleep-wakefulness cycles (Fig. 1a, b). In both sham and CCI mice, we applied optogenetic stimulation (473 nm, 1-ms pulse duration, 10 Hz × 1 s) to ChR2-expressing NAc neurons through a preinstall optical fiber. Through simultaneous single-unit recording, we screened and defined the in vivo ChR2-expressing neurons by 3 criteria: (1) they spiked consistently upon laser stimulation (>70% of firing rate), (2) they spiked with short latencies (1–3 ms) and low jitters (<3 ms) to rule out network implication, (3) laser-evoked spikes were consistent with the profile of spontaneous spikes (Fig. 1c, d, Supplementary Fig. 2c–e).

Mechanical allodynia, a type of evoked nociceptive response that is elicited by non-noxious mechanical stimulation, is the most common behavioral response in animals with neuropathic pain[35]. To test whether these CCI-activated ChR2-expressing NAc neurons were sensitive to non-noxious mechanical stimuli, we performed optrode recordings in c-Fos-tTA mice 3 weeks after CCI or sham surgery (Fig. 1a, e). Non-noxious stimuli (0.07-g von Frey filament stimuli), inducing nociceptive response in CCI mice but not sham mice, increased the firing rate in NAc neurons identified as CCI-activated (Fig. 1f–I, CCI-Identified, blue, $n = 28$ units from 6 mice), but not in the unidentified neurons from CCI mice (CCI-

Unidentified, cyan, $n = 17$ units from 6 mice) or NAc neurons from sham mice (Fig. 1I, Sham-Identified, black, $n = 34$ units from 7 mice; Group, $F_{(2,76)} = 50.35$, $p < 0.001$; before: CCI-IDed vs. Sh-IDed, $p < 0.001$; CCI-IDed vs. CCI-UI, $p < 0.001$; after: CCI-IDed vs. Sh-IDed, $p < 0.001$; CCI-IDed vs. CCI-UI, $p < 0.001$. Time, $F_{(1,76)} = 48.62$, $p < 0.001$; CCI-IDed: before vs. after, $p < 0.001$). These results show that CCI-activated NAc neurons exhibited increased activities upon nociceptive response.

Next, we detected that the CCI-activated, ChR2-expressing (CCI-ChR2) NAc neurons exhibited rhythmic activity changes during the sleep-wakefulness cycle. In CCI mice, the mean spiking rate of CCI-ChR2 neurons was considerably higher during wakefulness compared to either NREM or REM sleep (Fig. 1j) (Fig. 1k, CCI-IDed: State, $F_{(2,152)} = 39.05$, $p < 0.001$, wake vs. NREM, $p < 0.001$; wake vs. REM, $p < 0.001$). Furthermore, the spiking rate of these neurons in CCI mice was higher than CCI-Unidentified neurons or Sham-Identified neurons during wakefulness or REM sleep (Fig. 1k, Group, $F_{(2,76)} = 49.40$, $p < 0.001$; CCI-IDed vs. Sh-IDed: wake, $p < 0.001$, REM, $p < 0.001$; CCI-IDed vs. CCI-UI: wake, $p < 0.001$, REM $p < 0.001$). We also quantified the brain-state preference of recorded neurons by calculating their wake-NREM modulation [$(R_{wake} − R_{NREM})/(R_{wake} + R_{NREM})$, R is the averaged firing rate within each brain state] and REM-NREM modulation [$(R_{REM} − R_{NREM})/(R_{REM} + R_{NREM})$] (Fig. 1l). Compared with the identified neurons from sham mice and unidentified neurons from CCI mice, CCI-ChR2 NAc neurons showed increased wake-NREM ($F_{(2,76)} = 25.75$, $p < 0.001$, CCI-IDed vs. Sh-IDed: $p < 0.001$; CCI-IDed vs. CCI-UI: $p < 0.001$, One-way ANOVA test) and REM-NREM modulation ($F_{(2,76)} = 19.99$, $p < 0.001$, CCI-IDed vs. Sh-IDed: $p = 0.001$; CCI-IDed vs. CCI-UI: $p < 0.001$). Further analysis indicates that the activity of CCI-ChR2 NAc neurons was elevated during wakefulness. Specifically, the low spiking rate of these neurons during either NREM or REM sleep sharply increased when the mice transitioned to wakefulness, and vice versa (Fig. 1m, NREM→wake, $P < 0.001$, REM→wake, $P < 0.001$, Wilcoxon signed-rank test). This wakefulness-sleep-contingent activity change was not detected in randomly sampled NAc neurons in either sham or CCI mice (Supplementary Fig. 3a, b). Thus, the CCI-activated NAc neurons are preferentially activated during wakefulness and may account for the decreased NREM sleep and increased wakefulness in CCI-induced chronic pain conditions.

Based on the change features of neuronal activity upon regulating nociception and sleep/wakefulness, we proposed that the CCI-ChR2 NAc neurons form a dual-functional neuronal ensemble in CCI mice (called NAc ensemble in the following) that processes both pain and sleep/wakefulness information.

### Activation of NAc ensemble exacerbates pain-like behaviors and increases wakefulness

The above correlative results prompted us to explore the potential causal roles of NAc ensemble in regulating pain-like behaviors and sleep/wakefulness. In CCI mice (Fig. 2a, b, Supplementary Fig. 4a), we optogenetically stimulated (10 Hz) NAc ensemble neurons, which decreased both the PWL ($n = 14$ mice, $F_{(1.54, 20)} = 204$, $p < 0.0001$; BL vs. On, $p < 0.001$, On vs. Off, $p = 0.0002$, BL vs. Off, $p < 0.001$) and PWT ($n = 14$ mice, $F_{(2, 26)} = 117$, $p < 0.0001$; BL vs. On, $p < 0.001$, On vs. Off, $p < 0.0001$, BL vs. Off, $p < 0.001$) in CCI mice (Fig. 2c, d), but not in sham mice (Supplementary Fig. 5a, b), indicating that this ensemble regulated nociceptive responses underlying chronic pain conditions. On the other hand, optogenetic stimulation (10 Hz × 2 min) of the ensemble neurons triggered the transition from NREM sleep to wakefulness ($p < 0.05$, bootstrap), increased the percentage of wakefulness ($p < 0.0001$, bootstrap), and decreased NREM sleep ($p < 0.0001$, bootstrap), resulting in the overall promotion of wakefulness in CCI mice (Fig. 2e–g), but not in sham mice (Supplementary Fig. 5c–e). Moreover, compared with 5 Hz and 20 Hz blue laser stimulations, 10 Hz laser stimulations of NAc ensemble neurons had a greater effect on sleep-wake behavior (Supplementary Fig. 6a, b). Notably, both optogenetic activation and inactivation of NAc

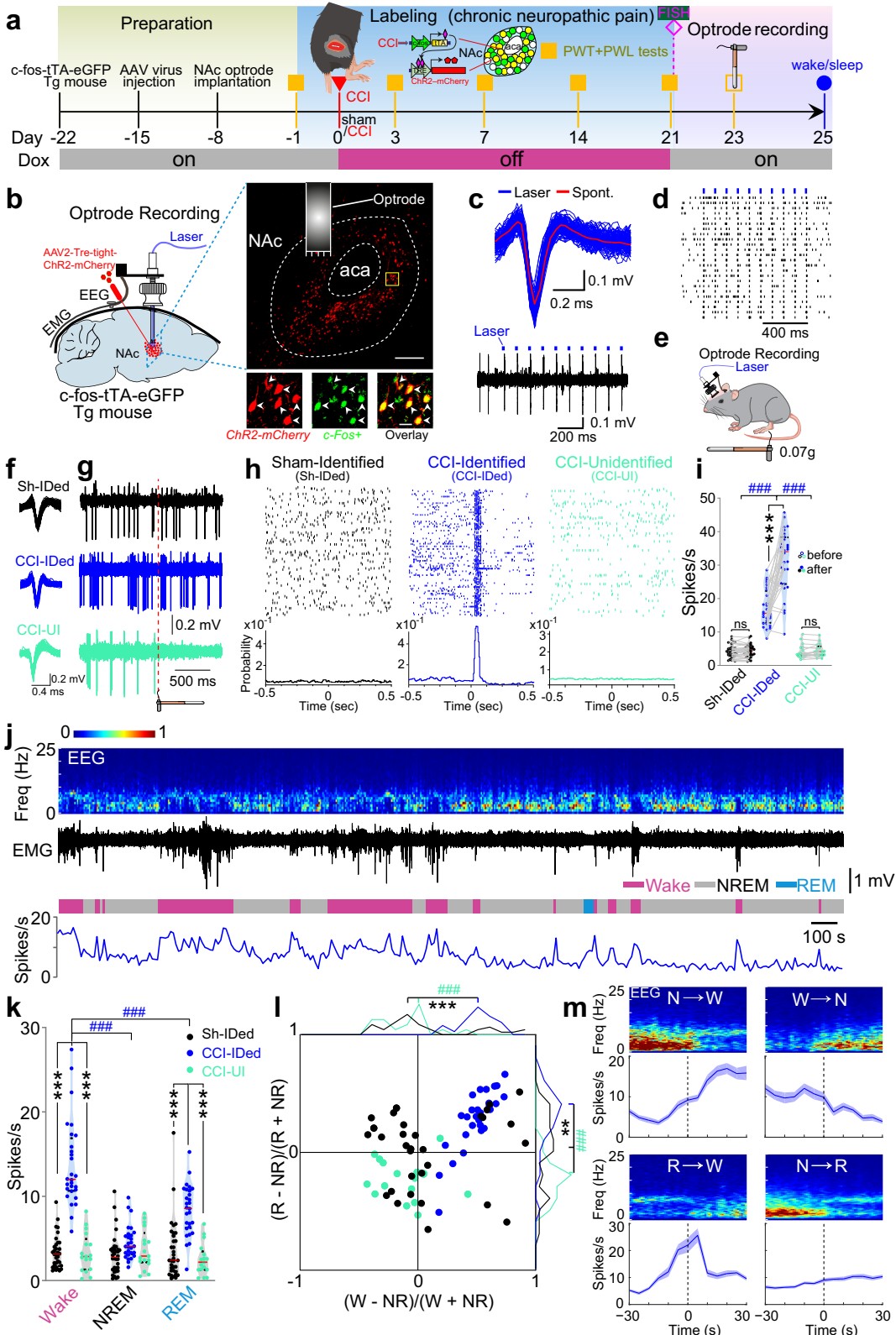

ensemble neurons did not affect the movement of paw withdrawal and general locomotion (Supplementary Fig. 7a–h).

We next asked whether direct sleep deprivation could mimic the effects of optogenetic stimulation of NAc ensemble to regulate the nociception of CCI mice. To address this question, we performed a 6-h sleep deprivation in CCI mice 21 days after CCI surgery (Supplementary Fig. 8a). This sleep deprivation exacerbated both mechanical

allodynia and thermal hyperalgesia in CCI mice (Supplementary Fig. 8b–c), with effects similar to that resulting from optogenetic activation of NAc ensemble.

Our subsequent results show that NAc non-ensemble neurons did not regulate sleep or nociception. Specifically, we used the "Tet-Off" and "Cre-Off" combination[36,37]. This involves injecting 2 AAVs (AAV-Tre3G-CRE-WPRE-pA and AAV-EF1a-DO-hChR2-mCherry-WPREs) into

**Fig. 1 | The dual-functioning NAc ensemble. a** Schematics showing the labeling and recording of NAc ensemble in c-fos-tTA transgenic mice underwent CCI or sham surgery. FISH (Fluorescent in situ Hybridization) was performed on a subgroup of CCI mice at day 21 after the CCI surgery. **b** Schematic configuration of multichannel optrode recording in the NAc from a CCI mouse and a representative segment showing the overlap of ChR2-mCherry and c-Fos+ neurons in the down insets. Brain figure adapted from Allen Mouse Brain Atlas (http://atlas.brain-map.org/). Scale bar: 200 μm (up) and 20 μm (down). **c** Waveforms of average spontaneous (red) and individual laser-evoked (blue) spikes from an identified neuron in the NAc ensemble. **d** Example of raster plot from a neuron in the NAc ensemble showing consecutive laser stimulation trials at 10 Hz. **e** Schematic depicting Frey filament stimulation and multichannel optrode recording. **f**–**h** Waveforms of spikes (**f**), representative traces (**g**), samples of peri-event raster plots (**h**) from an identified NAc neuron in a sham mouse (Sh-IDed), an identified neuron in the NAc ensemble (CCI-IDed), and an unidentified NAc neuron in a CCI mouse (CCI-UI) to

c-Fos-tTA mice two weeks before the CCI surgery, such that ChR2-mCherry was expressed selectively in c-Fos-negative neurons. After 21 days of CCI, we confirmed that ChR2-mCherry-expressing NAc neurons were c-Fos-negative, while c-Fos-positive neurons rarely expressed ChR2-mCherry ($1.36 \pm 0.37\%$ of c-Fos+ neurons expressed ChR2-mCherry and $2.60 \pm 0.27\%$ of ChR2-mCherry neurons were c-Fos+; Fig. 2a, h; Supplementary Fig. 4b). We operationally defined these ChR2-expressing neurons as NAc non-ensemble neurons. 10-Hz optogenetic stimulation of these non-ensemble neurons did not affect either nociception or sleep-wakefulness in CCI mice (Fig. 2i–m; Supplementary Fig. 4d). These results indicate that, in the NAc, the dual-functional ensemble is confined to pain-activated neurons.

### NAc ensemble formed in inflammatory pain
To further verify whether the NAc ensemble is formed under different chronic pain conditions, we examined NAc ensembles that were activated during the chronic inflammatory pain induced by complete Freund's adjuvant (CFA) (Fig. 3a). We applied two injections of CFA into the hind paw to establish a mouse model of chronic inflammatory pain (referred to as CFA mice). After 3–20 days post CFA injection, CFA mice displayed a reduction in the mechanical pain threshold, while control mice that received saline injections (referred to as saline mice) did not show this reduction (Fig. 3b). Twenty-four days post CFA injection, CFA mice exhibited the normal-like PWT compared with saline mice, suggesting a recovery from chronic inflammatory pain. We subsequently applied 10 Hz blue laser stimulations to activate the NAc ensemble. Consistently, activation of NAc ensemble significantly decreased PWT and NREM sleep and increased wakefulness in CFA mice, but not in sham mice (Fig. 3b–e). Next, to investigate whether the NAc ensemble induced by chronic inflammation pain also can be activated by sleep deprivation, we first labeled the NAc ensemble with mCherry using the Tet-Off system in c-fos-tTA mice received CFA injections. Subsequently, we applied sleep deprivation to CFA and saline mice in the absence of Dox. Double FISH was performed for c-Fos and mCherry (Fig. 3f). The majority of mCherry-tagged NAc ensemble neurons expressed c-Fos after sleep deprivation in CFA mice, while few mCherry-labeled neurons expressed c-Fos after sleep deprivation in Saline mice (Fig. 3g). Furthermore, behavioral tests showed that sleep deprivation re-induced a decrease in both mechanical PWT and thermal PWL in the recovered CFA mice from chronic inflammatory pain (Supplementary Fig. 9a–c). These findings dovetail with the CCI results, and provide evidence that the dual-functioning ensemble is a common circuit feature for multiple forms of pain.

### Inactivating NAc ensemble ameliorates pain-like behaviors and increases sleep
In contrast to activation, our subsequent results show that inactivation of NAc ensemble ameliorated pain-like behaviors and increased NREM sleep. Specifically, we used the same AAV but expressing eNpHR (Tre-

non-noxious stimuli. **i** Violin plot displaying the individual firing rates of Sh-IDed ($n = 34$ units from 7 mice), CCI-IDed $n = 28$ units from 6 mice), and CCI-UI ($n = 17$ units from 6 mice) before and after the non-noxious stimuli. In this and the following violin plots, data are presented as median (red line) with 25th and 75th percentile (dash line). **j** Representative firing rate of a neuron in the NAc ensemble (bottom) together with brain states (color-coded), EMG trace (middle), and EEG spectrogram (top). **k** Violin plot displaying the individual firing rates of the same three groups of neurons as (**i**) during different brain states. **l** The distributions of both Wake-NREM and REM-NREM modulations were significantly different among the neurons from the same three groups of neurons as (**i**, **k**). **m** Mean firing rates of identified NAc ensemble during different brain state transitions. Shading represents ± SEM. Two-way ANOVA and one-way ANOVA with Bonferroni's multiple comparisons test for (**i**, **k**) and (**l**), respectively. $^{###}p < 0.001$, $^{***}p < 0.001$, $^{**}p < 0.01$, ns, not significant.

tight-eNpHR-mCherry) in ensemble neurons in c-fos-tTA mice (Fig. 4a). After 21 days of CCI, optogenetic inhibition (589 nm) of the NAc ensemble restored the mechanical and thermal pain thresholds in CCI mice to baseline levels (Fig. 4b, c, Supplementary Fig. 10a, b). On the other hand, optogenetic inhibition of the NAc ensemble induced an abrupt increase in NREM sleep and, consequently, a reduction in wakefulness (Fig. 4d–f). Further analyses indicate that these changes were due to the increased wake→NREM and NREM→NREM transitions, as well as the decreased wake→wake and NREM→wake transitions in CCI mice (Fig. 4e and Supplementary Fig. 10d), but not sham mice (Supplementary Fig. 10c, e). Thus, with increased or decreased activities, the pain-defined NAc ensemble bidirectionally regulates nociception and sleep/wakefulness.

### Identification of cell types in the NAc ensemble
The NAc is mostly composed of two subtypes of principal neurons, one expressing dopamine D1, and the other expressing D2 receptors[38,39]. Using double FISH, we found that the majority of CCI-induced NAc ensemble expressed dopamine D1 receptor ($80.73 \pm 1.72\%$), and the minority of CCI-induced NAc ensemble expressed dopamine D2 receptor ($18.75 \pm 1.20\%$; D1 vs. D2, $p = 0.0079$, two-tailed Mann–Whitney test; Fig. 5a–c), indicating that D1 neurons comprised the majority of ensemble neurons.

To test the role of NAc D1 neurons in regulating nociception and sleep, we bidirectionally manipulated their activity. Previous study has shown that distinct frequency stimulations induced the frequency-dependent effects on cell firing in subthalamic nucleus and substantia nigra pars reticulata[40]. Specifically, subthalamic firing attenuated with ⩾20 Hz stimulation (silenced at 100 Hz), while substantia nigra pars reticulata decreased with ⩾30 Hz (silenced at 50 Hz)[40]. Consistently, low frequency stimulations could increase the firing rate of NAc D1R neurons (Supplementary Fig. 11a–c), while high frequency stimulations could inhibit the firing rate of these neurons (Supplementary Fig. 11d, e). We then optogenetically (473 nm, 10 Hz) activated D1 neurons in D1-Cre mice with Cre-dependent expression of ChR2 in the NAc. This activation decreased both the PWL (Fig. 5d, $n = 11$ mice, $F_{(2.143, 21.43)} = 55.14$, p < 0.0001, BL vs. Laser, $p < 0.001$, Pre vs. Laser, $p = 0.017$, Post vs. Laser, $p = 0.003$) and PWT (Fig. 5e, $n = 11$ mice, $F_{(1.930, 19.30)} = 25.71$, $p < 0.0001$ BL vs. Laser, $p < 0.001$, Pre vs. Laser, $p < 0.001$, Post vs. Laser, $p = 0.008$) in CCI mice. Moreover, optogenetic stimulation (10 Hz × 2 min) of NAc D1 neurons increased the percentage of wakefulness ($p < 0.0001$, bootstrap), and decreased NREM and REM sleep ($p < 0.0001$, bootstrap), resulting in an overall upregulation of wakefulness (Fig. 5f).

For optogenetic inhibition, we selectively expressed eNpHR in NAc D1 neurons. In CCI mice, yellow laser (589 nm, 8-s on/2-s off)-induced inhibition of NAc D1 neurons increased both the PWL and PWT to baseline levels (Fig. 5g, h). Meanwhile, this D1 neuron-selective inhibition also increased the percentage of NREM sleep ($p < 0.0001$, bootstrap) and decreased wakefulness ($p < 0.0001$, bootstrap, Fig. 5i).

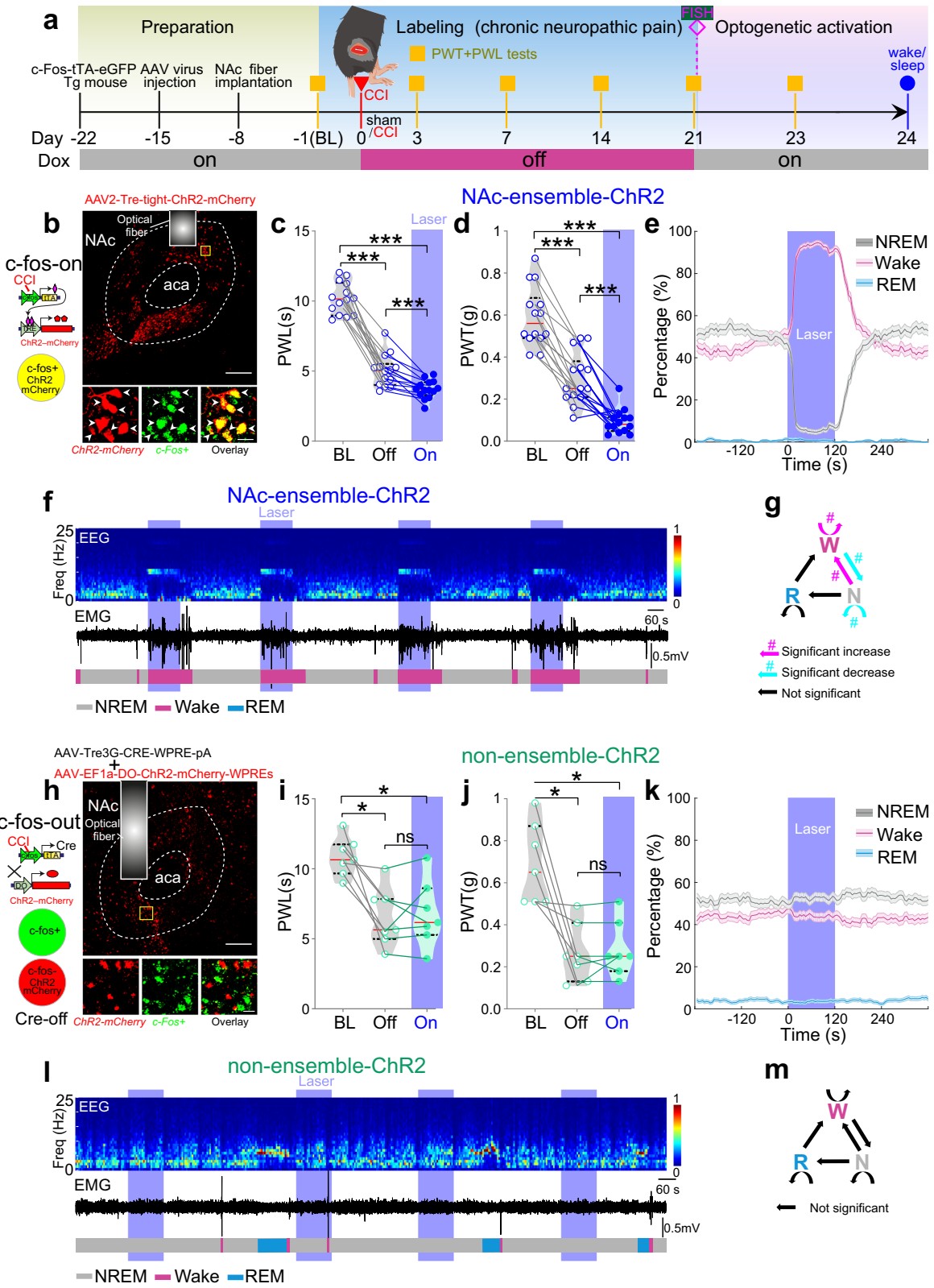

To test the role of NAc D2 neurons in regulating nociception and sleep, We then optogenetically (473 nm, 10 Hz) activated D2 neurons in D2-Cre mice with Cre-dependent expression of ChR2 in the NAc. This activation rescued both the PWL (Fig. 6a, $n = 13$ mice, $F_{(2.46, 29.5)} = 54.3$, $p < 0.0001$, BL vs. Laser, $p = 0.73$, Pre vs. Laser, $p < 0.001$, Post vs. Laser, $p < 0.001$) and PWT in CCI mice (Fig. 6b, $n = 13$ mice, $F_{(2.06, 24.8)} = 49.6$, $p < 0.0001$, BL vs. Laser, $p = 0.735$, Pre vs. Laser,

$p < 0.001$, Post vs. Laser, $p < 0.001$). Moreover, optogenetic stimulation (10 Hz × 2 min) of NAc D2 neurons increased the percentage of NREM sleep, and decreased wakefulness and REM sleep, resulting in an overall upregulation of NREM sleep (Fig. 6c). These results are consistent with previous studies[29,41] that activation of total NAc D2 neurons ameliorates pain-like behaviors and increases NREM sleep.

**Fig. 2 | Activation of NAc ensemble exacerbates pain-like behaviors and increases wakefulness. a** Similar to Fig. 1a, but with activation of NAc ensemble. **b** Left panel: the diagram for labeling c-Fos-positive cells with ChR2-mCherry. Right panel: Expression of ChR2-mCherry in NAc ensemble and the placement of optical fiber. Scale bar: top, 200 μm; bottom, 20 μm. **c**, **d** Optogenetic activation of NAc ensemble significantly decreases thermal PWL ($n = 14$ mice) and mechanical PWT ($n = 14$ mice). **e** Significant decrease in NREM sleep and increase in wakefulness, and no obvious change in REM sleep during laser stimulation, $n = 14$ mice. Shading represents ± SEM. **f** Representative EEG spectrogram (top), relative EMG trace (middle), and brain states (bottom) from a NAc-ensemble-ChR2 mouse during baseline conditions and blue laser activation of NAc ensemble. Blue stripe indicates laser stimulation (473 nm, 10 Hz, 120 s). **g** The diagram summarizes transitions that are significantly increased (magenta), decreased (cyan), or unaffected (black) by laser stimulation ($^\# p < 0.05$, bootstrap). **h** Left panel: the diagram for labeling c-Fos-negative cells with ChR2-mCherry. Right panel: Expression of ChR2-mCherry in non-ensemble and the placement of optical fiber. Scale bar: top, 200 μm; bottom, 20 μm. **i**, **j** No significant differences in PWL ($n = 7$ mice, $F_{(1.36, 8.14)} = 13.2$, $p = 0.0045$; BL vs. On, $p = 0.0392$, On vs. Off, $p > 0.99$, BL vs. Off, $p = 0.0146$) and PWT ($n = 7$ mice, $F_{(1.07, 6.41)} = 13.4$, $p = 0.0089$; BL vs. On, $p = 0.0356$, On vs. Off, $p > 0.99$, BL vs. Off, $p = 0.0256$) are observed in non-ensemble-ChR2 mice between with and without laser activations. **k** No effect of blue laser stimulation on brain states in non-ensemble-ChR2 mice ($n = 7$ mice, $p = 0.4989$, $= 0.1686$, and $= 0.105$ for NREM, wake, and REM states). **l** Similar to (**f**), but for activation of non-ensemble. **m** No effect of laser stimulation on transition probability between each pair of brain states in non-ensemble-ChR2 mice. Repeated measures one-way ANOVA test with Bonferroni's multiple comparisons test for (**c**, **d**, **i**, **j**). $^{***}p < 0.001$ and $^*p < 0.05$.

We next used the "Tet-Off" and "Cre-On" combination to specifically activate NAc-ensemble-D2 neurons. This involved injecting 2 AAVs (AAV-Tre3G-CRE-WPRE-pA and rAAV-D2-DIO-hChR2-mCherry-WPREs) into NAc of c-Fos-tTA mice two weeks before the CCI surgery, such that ChR2-mCherry was expressed selectively in c-Fos-D2 neurons (NAc-ensemble-D2 neurons) (Fig. 6d). These NAc-ensemble-D2 Neurons were preferentially active during wakefulness (Supplementary Fig. 12). Lastly, we found that 10-Hz optogenetic stimulation of NAc-ensemble-D2 neurons increased wakefulness and decreases NREM sleep (Fig. 6e, f). On the other hand, the same optogenetic stimulation of these NAc-ensemble-D2 neurons decreased in both the PWL and PWT in CCI mice after 21 days of CCI (Fig. 6g, h). These findings suggest that a small portion of NAc D2 neurons form the NAc ensemble.

Taken together, these results suggest that both D1 and D2 neurons are contributors to the NAc ensemble. These ensemble neurons consistently regulate both nociception and sleep.

## NAc ensemble neurons project to both POA and VTA

A possible mechanism for a single NAc ensemble to regulate two distinctly different physiological modalities may implicate their divergently projected downstream targets. We anterogradely traced the ensemble neurons in CCI mice, and detected an array of downstream brain regions that are known to regulate sleep/wakefulness or nociception, including the preoptic area (POA)[42–44], lateral hypothalamus area (LHA)[45–47], medial habenula (MHb)[48], ventral tegmental area (VTA)[24,49,50], supramammillary region (SuM)[51], ventrolateral periaqueductal gray (vlPAG)[52], dorsal raphe (DR)[53], and locus coeruleus (LC)[54] (Fig. 7a–c). FISH double-labeling detected *Vgat* expression in ensemble neurons, suggesting that they were principal medium spiny neurons and influenced downstream brain regions through GABAergic transmission (Fig. 7d). It has been proposed that activation of GABAergic neurons could reduce sleep by inhibiting sleep-promoting neurons[55–57] and exacerbate pain-like behaviors by inactivation of pain-relieving neurons[58,59].

Among the downstream projection regions of ensemble neurons, both the POA[42–44,60–62] and VTA[50,63–65] are enriched in neuronal types that regulate sleep and nociception. We used rAAV2-retro to express mCherry versus eGFP in projection fibers within the POA versus VTA, respectively, which was expected to retrogradely label projection neurons in CCI mice[66] (Fig. 7e–g). Despite certain degrees of separation, mCherry versus eGFP signals were overlapped in ~30% (31.86 ± 2.52%) of c-Fos+ NAc ensemble neurons (Fig. 7g, h). Thus, a portion of NAc ensemble neurons divergently projected to both POA and VTA. Are the POA and VTA the downstream targets through which the NAc ensemble achieve the modality specificity? We tested this below.

## Inhibition of NAc-ensemble-innervated POA or VTA neurons

To map the relevant downstream circuit, the conventional method relies on activation of ChR2-expressing axon terminals in a given

target nucleus[67]. Therefore, we direclty activated the axon terminals of NAc ensemble in POA and VTA (Supplementary Fig. 13a–f). We found that either activation of POA or VTA projections innervated by NAc ensemble reduces both pain thresholds and NREM sleep (Supplementary Fig. 13a–f). It has been found that terminal activation could result in unwanted activation of collateral targets via antidromic stimulation[67]. To test that, we did tetrode recording at the soma of NAc neurons and POA neurons, simultaneously applied laser activations of NAc neurons terminals in VTA (Supplementary Fig. 13g, h). We found "antidromic spikes" in the soma of NAc neurons (Supplementary Fig. 13g). POA neurons were inhibited during laser stimulations (Supplementary Fig. 13h). We speculated that activating NAc-ensemble-POA or NAc-ensemble-VTA projections, respectively, could cause "antidromic spikes" in collateral targets (VTA and POA), eventually, had similar effects on nociceptive responses and sleep-wakefulness states.

Therefore, to avoid "antidromic spikes" and selectively inhibit ensemble-innervated POA or VTA neurons, we used a dual-viral system. We injected the first, anterograde trans-synaptic AAV (pAAV1-PTRE-tight-NLS-Cre) into the NAc, and one week later injected another AAV (AAV8-Ef1a-DIO-eNpHR-mCherry) into the POA or VTA, respectively, allowing the expression of eNpHR selectively in the POA or VTA neurons that were innervated by c-Fos-positive NAc neurons in CCI mice (Fig. 8a, b, h). Moreover, we injected AAV8-Ef1a-DIO-eNpHR-mCherry only into POA and VTA, respectively, without injection of pAAV1-PTRE-tight-NLS-Cre into NAc. We find no mCherry expression in POA and VTA in the absence of Cre (Supplementary Fig. 14a, b).

Five weeks later, optogenetic inhibition (Fig. 8c, 589 nm, 8-s on/2-s off) of these POA neurons did not change the nociceptive threshold in both PWL (Fig. 8d, $n = 14$ mice, $F_{(1.977, 25.70)} = 56.58$, $p < 0.001$, Pre vs. Laser, $p > 0.99$, Post vs. Laser, $p > 0.99$, Pre vs. Post, $p > 0.99$, BL vs. Laser, $p < 0.001$, BL vs. Pre, $p < 0.001$, BL vs. Post, $p < 0.001$) and PWT (Fig. 8e, $n = 14$ mice, $F_{(2.485, 32.30)} = 28.79$, $p < 0.001$, Pre vs. Laser, $p > 0.99$, Post vs. Laser, $p > 0.99$, BL vs. Laser, $p < 0.001$, BL vs. Pre, $p < 0.001$, BL vs. Post, $p < 0.001$). However, similar optogenetic inhibition of these POA neurons induced an immediate transition from NREM sleep to wakefulness (Fig. 8f, g). Moreover, the same optogenetic stimulations of the POA neurons innervated by the identified NAc neurons in sham mice did not change both the nociceptive responses and sleep/wakefulness rhythm (Supplementary Fig. 14c–e).

On the other hand, in CCI mice in which eNpHR-mCherry was selectively expressed in VTA neurons that were innervated by the NAc ensemble (Fig. 8a, h), optogenetic inhibition (Fig. 8i) decreased the threshold of nociceptive response measured by PWL (Fig. 8j, $n = 13$ mice, $F_{(2.051, 24.61)} = 113.7$, $p < 0.001$, Pre vs. Laser, $p < 0.001$, Post vs. Laser, $p < 0.001$, BL vs. Laser, $p < 0.001$, BL vs. Pre, $p < 0.001$, BL vs. Post, $p < 0.001$) and PWT (Fig. 8k, $n = 13$ mice, $F_{(2.097, 25.17)} = 42.22$, $p < 0.001$, Pre vs. Laser, $p = 0.01$, Post vs. Laser, $p = 0.003$, BL vs. Laser, $p < 0.001$, BL vs. Pre, $p < 0.001$, BL vs. Post, $p < 0.001$), but did not affect the sleep/wakefulness rhythm (Fig. 8l, m).

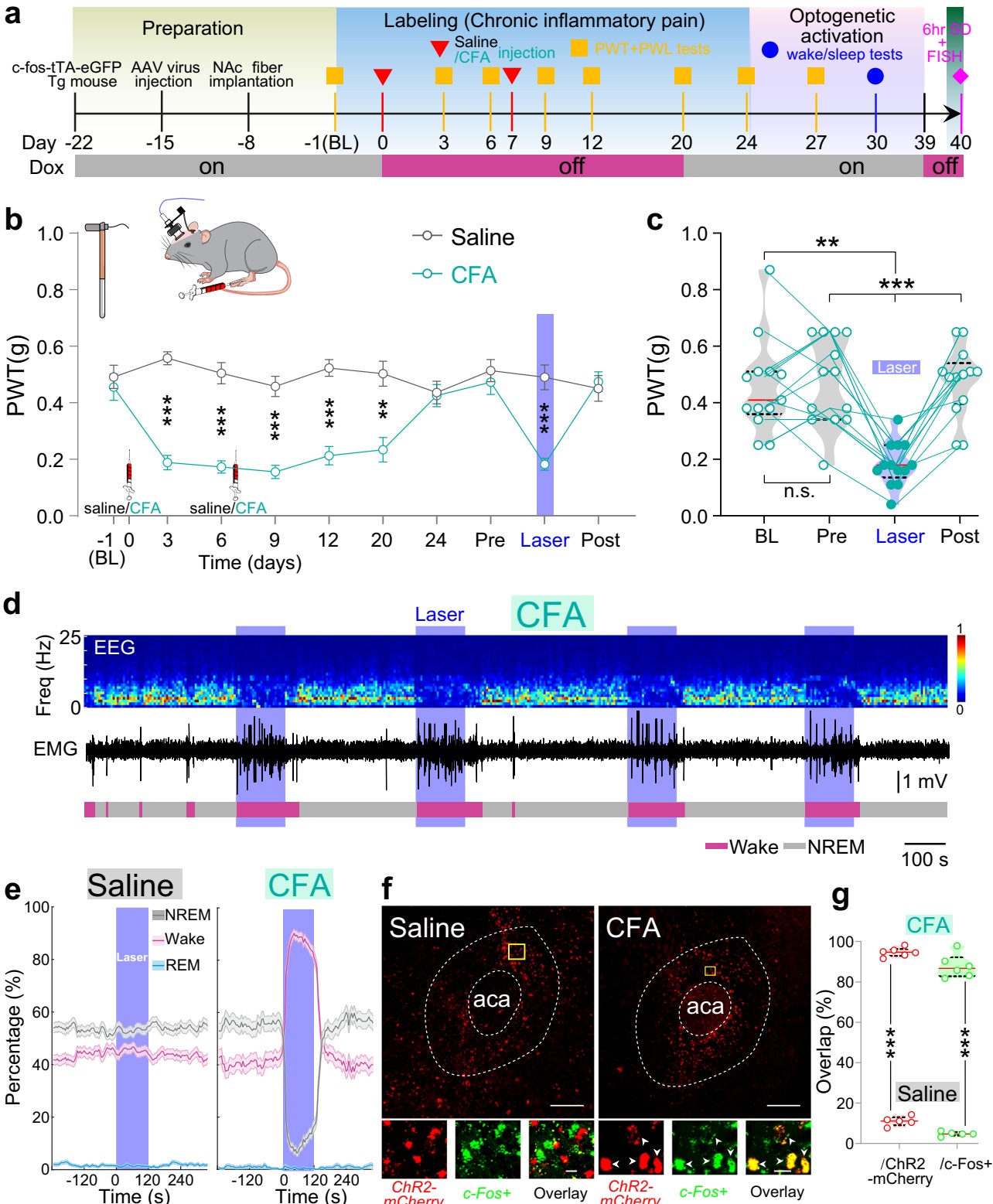

Moreover, the same inhibitions of the VTA neurons innervated by the identified NAc neurons in sham mice did not change both the nociceptive thresholds and sleep/wakefulness states (Supplementary Fig. 14f–h).

These results taken together indicate POA and VTA neurons as two downstream targets that diverge the regulatory function of NAc ensemble toward nociception versus sleep/wakefulness, respectively.

## Discussion

Since the reembrace of the Engram Hypothesis, extensive results support that engrams or ensembles are functionally distinct, each encoding a specific behavior[17–20]. However, studies of generalization or relatively sophisticated behaviors show that, under certain circumstances, ensemble neurons can be multi-functional, simultaneously encoding different aspects of behaviors[21,22]. Our current results suggest that the same ensemble neurons encode information of both

**Fig. 3 | NAc ensemble formed in inflammatory pain. a** Similar to Fig. 2a, but applying CAF injections. Sleep deprivation (SD) and FISH were performed at the end of experiment. **b** Compared with the saline mice (black, $n = 13$ mice), CFA mice (cyan, $n = 13$ mice) exhibited significantly decreased PWT at days 3, 6, 9, 12, 20 following CFA injection, and during laser stimulation (Group, $F_{(1,24)} = 54.51$, $p < 0.001$; saline vs. CFA, days 3: $p < 0.001$, days 6: $p < 0.001$, days 9: $p < 0.001$, days 12: $p < 0.001$, days 20: $p = 0.002$, during laser $p < 0.001$). Data are mean ± SEM. BL, baseline; Pre, pre-laser stimulation; Post, post-laser stimulation; **c** Compared with baseline PWT, optogenetic activation of NAc ensemble significantly decreases PWT ($n = 13$ mice, $F_{(2.13, 25.6)} = 17.7$, $p < 0.001$; BL vs. Laser, $p = 0.0022$, Pre vs. Laser, $p = 0.0002$, Post vs. Laser, $p < 0.001$). **d** Similar to Fig. 2f, but in CFA mice. **e** Percentage of time in different brain states before, during, and after 10 Hz (120 s) laser stimulation in saline (left, $n = 13$ mice) and CFA mice (right, $n = 11$ mice). Note no

significant difference in the brain states of saline mice ($p = 0.163$, = 0.257, and = 0.363 for NREM, wake, and REM states), but the significant decrease in NREM sleep ($p < 0.0001$, bootstrap) and increase in wakefulness ($p < 0.0001$), and no obvious change in REM sleep during laser stimulation ($p = 0.110$) in CFA mice. Shading represents ± SEM. **f** Fluorescence image showing double FISH of ChR2-mCherry (red) and c-Fos (green) in saline (left) and CFA mice (right). Arrowheads indicate co-labeled neurons. Scale bar: top, 200 μm; bottom, 15 μm. **g** Statistical analysis of the overlap between ChR2-mCherry and c-Fos expressing neurons in CFA mice ($n = 6$ mice) and saline mice. **$p = 0.0043$ versus the overlap percentages in saline mice (two-tailed Mann–Whitney test). Two-way ANOVA test and repeated measures one-way ANOVA test with Bonferroni's multiple comparisons test for (**b**) and (**c**), respectively. ***$p < 0.001$ and **$p < 0.01$.

nociception and wakefulness/sleep, providing a conceptual leap that a single neuronal ensemble encodes two physiological responses of distinct modalities. These findings not only provide new insights into therapeutic strategies for pain and sleep disorder, but also raise several questions for a conceptualization of the ensemble function.

Chronic pain and sleep disturbance are frequently encountered clinically[12,13]. Echoing these clinical observations, results from our current mouse model showed that pain conditions and sleep abnormalities aggravated each other in CCI mice (Supplementary Figs. 1, 8, and 9). However, the neural basis substantiating this comorbidity remains poorly understood, preventing effective clinical intervention. The NAc is involved in the regulation of both nociception[68,69] and sleep-wakefulness[15,26]. Notably, there are wake-promoting[26] and pain-exacerbating cell types in the NAc[27]. Our results show that some of these sleep-regulating NAc neurons also regulate pain experience.

Combining optrode recording, viral tracing, FISH, and optogenetic manipulation, we were able to identify, characterize, and manipulate a dual-functioning NAc ensemble. We labeled a dual-functioning NAc ensemble which was pain- and sleep-associated between day 0 and 21 after CCI surgery. The identified neurons in sham mice may be activated and labelled by other behaviors (such as feeding, social interaction, etc.) during this period after sham surgery. Our results show that neurons in the dual-functioning ensemble in CCI mice exhibited several cellular dynamics distinct from non-ensemble neurons and the identified neurons in sham mice (Fig. 1f–i, k–m, and Supplementary Fig. 3). Although non-ensemble neurons in CCI mice and the identified neurons in sham mice showed similar mean firing rates during awake, NREM, and REM states (Fig. 1k), non-ensemble neurons and the identified neurons in sham mice exhibited a higher degree of functional diversity (Fig. 1l), with minimal changes in firing properties upon either brain state transition (Supplementary Fig. 3) or non-noxious stimuli (Fig. 1f–i). Importantly, bidirectional optogenetic manipulations of non-ensemble neurons in CCI mice (Supplementary Fig. 4) and the identified neurons in sham mice (Supplementary Figs. 5, 10, 14) did not affect sleep-wakefulness or pain response. Thus, non-ensemble neurons in CCI mice and the identified neurons in sham mice might participate in the processing of feeding, socialization, cognition, emotion, action, and other physiological modalities that require the NAc[70]. The manipulations of the non-ensemble neurons in CCI mice and the identified neurons in sham mice might affect other behavioral responses. On the other hand, the number of NAc non-ensembles neurons was relatively less compared with NAc ensemble neurons (Fig. 2b, h). It is also possible that activation of non-ensembles neurons (a small subset of neurons) might not affect sleep-wakefulness or pain response.

Double FISH indicates that NAc ensemble consists primarily of D1 neurons and partially of D2 neurons (Fig. 5). Previous study has found that NAc D1/D2 neurons are important not only for sleep-wakefulness, but also for nociception[26,29,41]. Our finding demonstrates the differences in regulating pain-like behaviors and wakefulness between NAc-

ensemble-D2 neurons and total NAc-D2 populations. Activating NAc-ensemble-D2 neurons exacerbates nociceptive responses and reduces NREM sleep, while activation of total NAc-D2 populations produces the opposite effects (Fig. 6). Therefore, it will be interesting to do molecular marker analyses on NAc-D2 neurons to investigate whether different markers exist between D2-ensemble and D2-non-ensemble neurons in the future.

Chronic pain and sleep disturbance reciprocally aggravate each other such that pain induces sleep disturbance while sleep disturbance lowers pain thresholds and increases spontaneous pain[11]. The downstream circuit targets of the NAc ensemble, including the POA, LHA, VTA, SuM, vlPAG, DR, and LC, play differential roles in regulating sleep-wakefulness and modulation of pain sensitivity[24,42–47,49,51–54,60,62,64,65,71] (Fig. 7). Among these downstream regions, the POA contains several sleep-promoting cell types and serves as a hub for sleep regulation[42–44]. In parallel, the POA also plays a role in the regulation of pain sensitivity[60–62]. Similarly, the VTA contains both sleep-promoting neurons[63] and pain-relieving neurons[64,65]. Retrograde tracing provided further evidence that NAc ensemble simultaneously sent monosynaptic inputs onto both POA and VTA neurons (Fig. 7e–g). Due to the antidromic stimulation[67], activations of POA or VTA projections innervated by NAc ensemble were different from transsynaptic cell body stimulations in POA or VTA. Either activating POA or VTA projections reduced both pain thresholds and NREM sleep (Supplementary Fig. 13a–f). However, inhibition of ensemble-innervated POA neurons selectively impaired NREM sleep without affecting chronic neuropathic pain sensitivity (Fig. 8), while inhibition of ensemble-innervated VTA neurons selectively exacerbated mechanical allodynia and thermal hyperalgesia without affecting sleep (Fig. 8). All these results also suggested that decreased PWL and PWT (Figs. 2, 3, 5, 6, 8) were not resulting from increased arousal. Moreover, the optogenetic inhibitions of the POA or VTA neurons innervated by the identified NAc neurons in sham mice did not change the nociceptive thresholds and sleep/wakefulness states (Supplementary Fig. 14), and might affect other behaviors. These effects of ensemble-innervated POA or VTA neurons in CCI mice, taken together with the effect of optogenetic activation/inhibition of the NAc ensemble, may reflect an inhibition-dominant regulation model. Specifically, activated NAc ensemble neurons provide GABAergic inhibition of POA and VTA neurons, resulting in decreased activities of the ensemble-innervated neurons that contribute to behavioral regulation. This scenario may explain why chronic pain and sleep disturbance are mutually interacting, each increasing the risk for the emergence and/or exacerbation of the other, and this may lead to a vicious cycle.

In addition to the NAc→POA and NAc→VTA projections, other potential pathways may also contribute to sleep-pain regulation. Therefore, further studies that examine the potential roles of specific connections from the NAc ensemble to other brain areas in reducing sleep and exacerbating pain-like behaviors are warranted.

Patients with comorbid chronic pain and sleep disturbances have greater severity and longer duration of both symptoms. Since some medications are only effective for one symptom[11,72], pharmacological

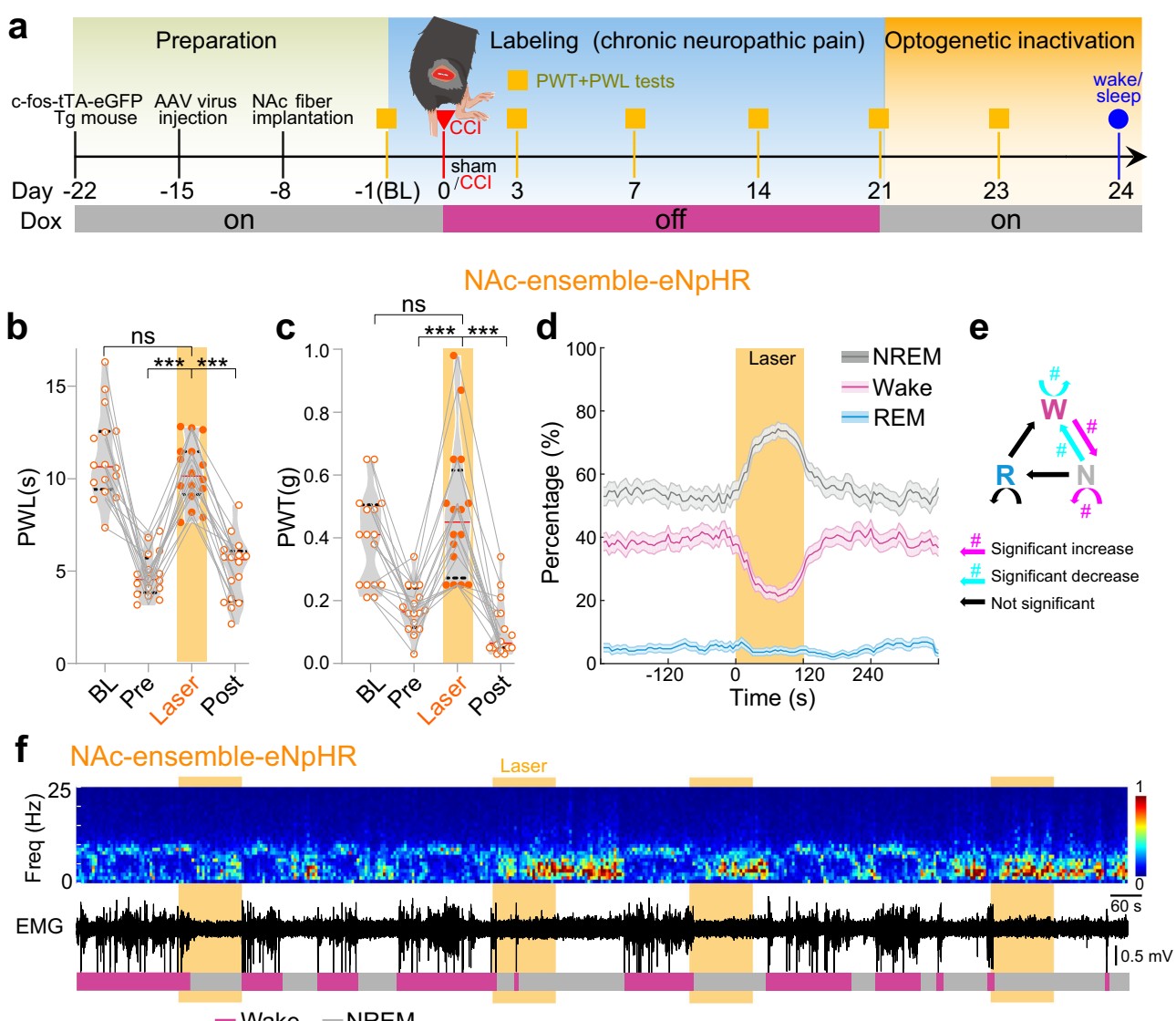

**Fig. 4 | Inactivating NAc ensemble ameliorates pain-like behaviors and increases sleep. a** Experimental scheme showing the labeling and inactivation of NAc ensemble in c-Fos-tTA transgenic mice underwent CCI or sham surgery. **b, c** Optogenetic inactivation (589 nm, yellow laser, 8 s on/2 s off, 120 s) of NAc ensemble significantly increased thermal PWL ($n = 16$ mice, $F_{(1.55, 23.3)} = 54.3$, $p < 0.001$, Repeated one-way measures ANOVA test; BL vs. Laser, $p > 0.99$, Pre vs. Laser, $p < 0.001$, Post vs. Laser, $p < 0.001$, Bonferroni's multiple comparisons test) and mechanical PWT ($n = 16$ mice, $F_{(2.03, 30.5)} = 32.50$, $p < 0.001$, Repeated one-way measures ANOVA test; BL vs. Laser, $p = 0.96$, Pre vs. Laser, $p < 0.001$, Post vs. Laser, $p < 0.001$, Bonferroni's multiple comparisons test) to the baseline level in NAc ensemble-eNpHR mice. BL, baseline; Pre, pre-laser stimulation; Post, post-laser stimulation. **d** Percentage of time in different brain states before, during, and after yellow laser inactivation of NAc ensemble (589 nm, yellow laser, 8 s on/2 s off, 120 s;

$n = 11$ mice). Note the significant increase in NREM sleep ($p < 0.0001$, bootstrap) and decrease in wakefulness ($p < 0.0001$, bootstrap), and no obvious change in REM sleep during laser stimulation ($p = 0.242$, bootstrap). Shading represents ± SEM. **e** The changes of transition probability between each pair of brain states in NAc ensemble-eNpHR mice during yellow laser stimulation. The diagram summarizes transitions that are significantly increased (magenta), decreased (cyan), or unaffected (black) by laser stimulation. Magenta and cyan asterisk (#) indicates significant increase and decrease in transition probability during laser stimulation compared to baseline ($p < 0.05$, bootstrap). **f** Representative EEG spectrogram (top), relative EMG trace (middle), and brain states (bottom) from a NAc ensemble-eNpHR mouse during baseline conditions and yellow laser inactivation of NAc ensemble. Yellow stripe indicates laser stimulation (589 nm, yellow laser, 8 s on/2 s off, 120 s). Freq. represents frequency. ***$p < 0.001$, ns not significant.

options for the comorbidity between chronic pain and sleep disturbance remain limited. Our current findings set the NAc ensemble as an interplay regulating both sleep and pain-like behaviors, raising the possibility of developing NAc-centered treatments, such as deep brain stimulation or transcranial magnetic stimulation[73–77]. Alternatively, future work for identifying small molecules that can target NAc ensemble neurons could be developed for treating the comorbidity.

Our current study is focused on the NAc, which is only a part of the circuit representation of the pain-sleep interaction in the brain. Our results do not rule out other brain areas that may also exhibit ensemble-like patterns of circuit dynamics that regulate pain-like

behaviors and sleep. Indeed, our demonstration of the dual-functioning ensemble may provide a conceptual standpoint to explore the common circuit underpinnings that mediate the interaction of behaviors of different modalities.

## Methods
### Animals
The care and use of animals and the experimental protocols (202207S001) used in this study were approved by the Institutional Animal Care and Use Committee and the Office of Laboratory Animal Resources of Xuzhou Medical University under the Regulations for the

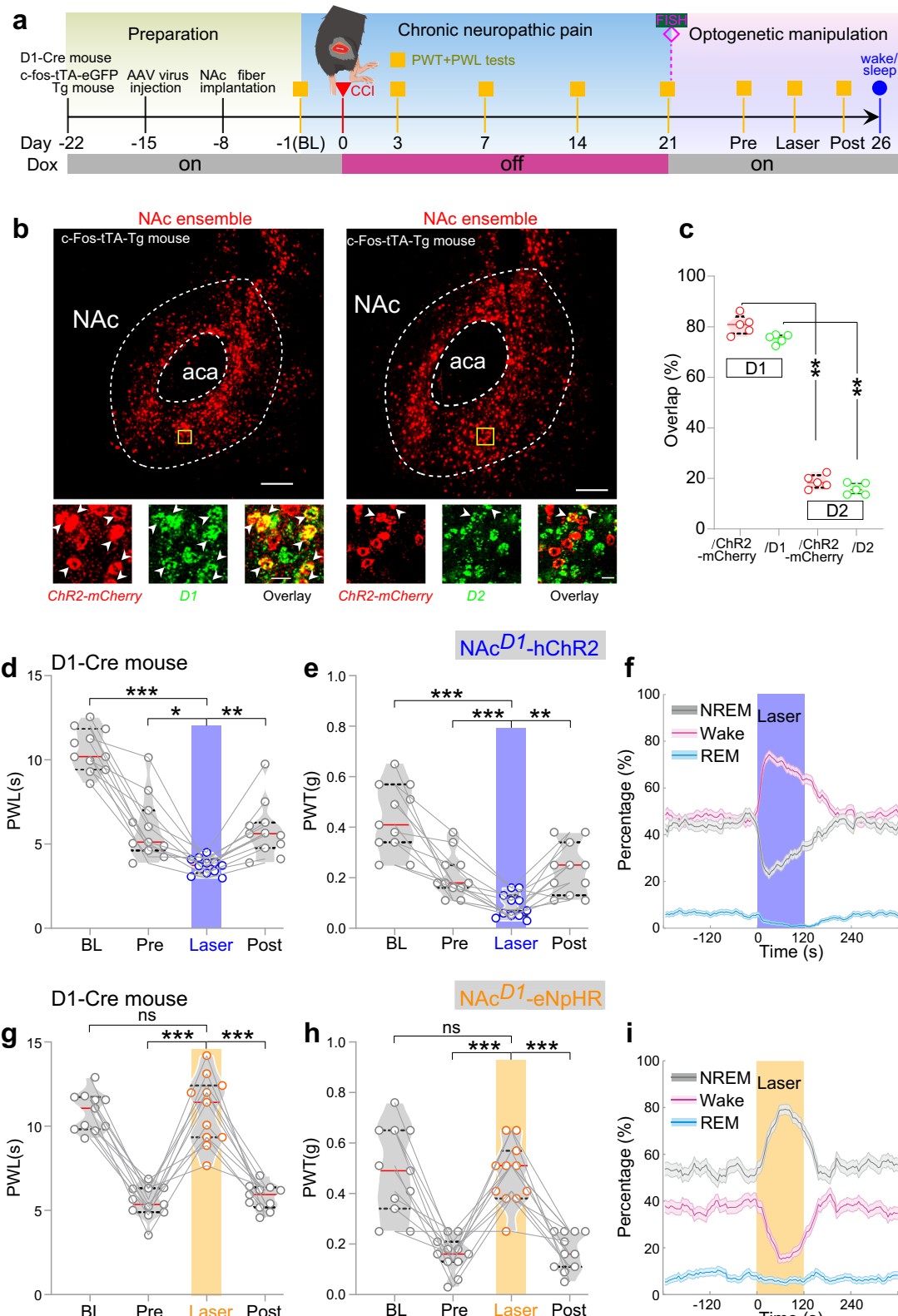

Administration of Affairs Concerning Experimental Animals (1988) in China. Male and female c-fos-tTA mice (stock number: O18306) were obtained from The Jackson Laboratory. Male and female D1R-Cre ((B6.FVB(Cg)-Tg(Drd1a-Cre) EY266Gsat/Mmucd) and D2R-Cre (B6.FVB(Cg)-Tg(Drd2-cre) ER44Gsat/Mmucd) mice were obtained from the GENSAT. Male and female C57/BL6J mice were obtained from the Laboratory Animal Center at Xuzhou Medical University. All mice

(3–5 months old) were housed in an environment with a constant temperature of $22 \pm 1\,°C$, humidity of $50 \pm 1\%$, and under12 hr light/12 hr dark cycle (light on from 7 a.m. to 7 p.m.) with ad libitum food and water.

## Mouse model of neuropathic pain
Chronic constrictive injury of the sciatic nerve (CCI) was performed to establish a neuropathic pain model[78]. Briefly, after anesthesia induction,

**Fig. 5 | D1 neurons form the NAc ensemble. a** Similar to Fig. 2a, but with activation of NAc D1 neurons in D1-cre mice. **b** Expression of ChR2-mCherry in NAc ensemble. Fluorescence image showing the double FISH of ChR2-mCherry (red) and dopamine D1 (left) and D2 (right) receptor (green) in c-Fos-tTA transgenic mice. Arrowheads indicate co-labeled neurons. Scale bar: top, 200 μm; bottom, 20 μm. **c** Statistical analysis of the overlap between ChR2-mCherry and dopamine D1/D2 receptor expressing neurons in c-Fos-tTA transgenic mice ($n = 5$ mice). Data are presented as median (red line) with 25th and 75th percentile (black dash line) (**$p = 0.0079$, two-tailed Mann–Whitney test). **d, e** Optogenetic activation (473 nm, 10 Hz) of NAc D1 neurons significantly decreases thermal PWL ($n = 11$ mice) and mechanical PWT ($n = 11$ mice) in D1-cre mice. **f** Optogenetic activation of NAc D1 neurons (473 nm, 10 Hz, 120 s) significantly increase wakefulness ($n = 11$ mice,

$p < 0.0001$, bootstrap) and decrease NREM ($p < 0.0001$) and REM ($p < 0.0001$) sleep. Data are mean ± SEM. Shading represents ± SEM. **g, h** Optogenetic inactivation of NAc D1 neurons rescues pain thresholds in PWL ($n = 11$ mice, $F_{(2.012, 20.12)} = 60.70$, $p < 0.001$, BL vs. Laser, $p > 0.99$, Pre vs. Laser, $p < 0.001$, Post vs. Laser, $p < 0.001$) and PWT ($n = 11$ mice, $F_{(2.046, 20.46)} = 23.20$, $p < 0.001$, BL vs. Laser, $p > 0.99$, Pre vs. Laser, $p < 0.001$, Post vs. Laser, $p < 0.001$) in D1-cre mice underwent CCI surgery. **i** Optogenetic inactivation of NAc D1 neurons (589 nm, yellow laser, 8 s on/2 s off, 120 s) significantly increase NREM sleep ($n = 11$ mice, $p < 0.0001$, bootstrap) and decrease wakefulness ($p < 0.0001$), and do not affect REM sleep ($p = 0.173$). Shading represents ± SEM. Repeated measures one-way ANOVA test with Bonferroni's multiple comparisons test for (**d, e, g, h**). *$p < 0.05$, **$p < 0.01$, ***$p < 0.001$, ns not significant.

the right common sciatic nerve of each mouse was exposed at the midthigh level. Three ligatures (non-absorbable, 4-0 silks) were tied loosely around it with ~1 mm between ligatures. Mice in the sham group received surgery identical to that described in CCI but without such constrictive injury.

## Mouse model of inflammatory pain

Complete Freund's adjuvant (CFA, 10 μl, Sigma-Aldrich) was injected unilaterally into the intraplantar surface of the hind paw in mice using a 20 μl Hamilton syringe under anesthesia, whereas control mice were injected with 0.9% saline. The persistence of inflammatory pain was ensured by a second CFA (10 μl) injection on the 7th day.

## Virus

AAV2-Tre-tight-ChR2-mCherry, AAV2-Tre-tight-eNpHR-mCherry, AAV2-Tre-tight-ChR2-eGFP, rAAV2-retro-hSyn-mCherry, rAAV2-retro-hSyn-eGFP, pAAV1-PTRE-tight-NLS-Cre, pAAV-EF1α-DIO-hChR2(H134R)-eYFP, and AAV8-Ef1a-DIO-eNpHR-mCherry were purchased from OBIO (Shanghai, China). AAV-Tre3G-CRE-WPRE-pA and AAV-EF1a-DO-ChR2-mCherry-WPREs, and rAAV-D2-DIO-hChR2-mCherry-WPREs were purchased from BrainVTA (Wuhan, China). All viral vectors were aliquoted and stored at −80 °C until use.

## Fluorescent in situ hybridization (FISH)

Mice were deeply anesthetized with sodium pentobarbital (100 mg/kg), perfused transcardially with ice-cold 0.1 M phosphate-buffered saline (PBS), and followed by 4% paraformaldehyde (PFA) in PBS. The mouse brains were carefully dissected out from the skull and post-fixed 24 h by immersion in 4% PFA at 4 °C. Then, the fixed brains were dehydrated with 30% sucrose in PBS at 4 °C for 48 h. After embedding and freezing, brains were sectioned into 20 (for FISH) or 50 μm coronal slices using a cryostat. RNAscope in situ hybridization was performed according to the manufacturer's instructions (Advanced Cell Diagnostics).

The sections were imaged with a confocal microscope controlled by Zen2 software (LSM 880, Zeiss, Oberkochen, Germany) and the images were processed with Fiji-ImageJ (V_1.52a).

## Surgery

Prior to the surgery, C57/BL6J or c-fos-tTA mice were anesthetized with 1% sodium pentobarbital (40 mg/kg, intraperitoneal injection, i.p.) and placed in a stereotaxic frame (RWD Life Technology Co. Ltd., Shenzhen, China). Body temperature was kept stable with a heating pad at 37–38 °C throughout the surgery. Ophthalmic ointment was applied in the eyes of mice to prevent drying. After the surgery, mice were placed in a cage with a heating pad underneath, and returned to their home cage when fully awake.

## EEG/EMG recording electrodes, optrodes, tetrodes, and optical fibers implantations

EEG/EMG recording electrodes, optrodes, tetrodes, and optical fibers implantations were performed in anesthetized mice[34]. For EEG implantation, two stainless steel screws were driven into the skull

(bregma: AP = −3.5 mm; ML = ± 3.0 mm). EMG implantation for recording sleep-wake state, two Teflon-coated annealed stainless steel wires were inserted into the neck musculature. EMG implantation for recording CCI hindlimb movement, two Teflon-coated annealed stainless steel wires were implanted on the medial gastrocnemius muscles[79]. A skin incision was made on the dorsal side of the mouse. The two wires were placed into MG belly and looped around MG cauda. The EEG and EMG electrodes were then soldered to a connector. Next, custom-made optrodes, tetrodes, or optic fibers were implanted into NAc, POA, or VTA at the same location of virus injection. The optrodes or optic fibers were secured to the skull together with EEG and EMG electrodes using dental cement. After surgery, the mice were allowed to recover for 1 week before the experiments.

## Virus injection

AAV virus was injected using a syringe nanoliter infusion pump (ZS, Dichuang, Beijing, China) and a 10 μl Hamilton syringe (~350 nl at a rate of 100 nl/min), followed by a 5-min pause to minimize backflow. AAV virus was injected into the side contralateral to CCI/CFA (paw). For optogenetic manipulation of NAc ensemble, AAV2-Tre-tight-ChR2-mCherry or AAV2-Tre-tight-eNpHR-mCherry was injected into NAc (coordinates, bregma: AP = +1.18 mm; ML = +1.13 mm; DV = −3.7 mm). For optogenetic activation of non-ensemble, we injected 1:1 volume mixtures of AAV-Tre3G-CRE-WPRE-pA and AAV-EF1a-DO-ChR2-mCherry-WPREs into NAc. For retrograde tracing, rAAV2-retro-hSyn-mCherry and rAAV2-retro-hSyn-eGFP were injected into POA (bregma: AP = +0.14 mm; ML = +0.8 mm; DV = −5.2 mm) and VTA (bregma: AP = −3.4 mm; ML = +0.5 mm; DV = −4.1 mm), respectively. For optogenetic inactivation of NAc ensemble→POA and NAc ensemble→VTA neurons, pAAV1-PTRE-tight-NLS-Cre was firstly injected into NAc and one week later AAV8-Ef1a-DIO-eNpHR-mCherry was injected into POA and VTA, respectively. Erythromycin ointment was applied locally to avoid infection.

## NAc ensemble labeling

For NAc ensemble labeling, we used c-Fos-tTA transgenic mice combined with the TRE system and conducted CCI surgery or CFA injection. We unilaterally injected these mice with adeno-associated virus 2 (AAV2-Tre-tight-ChR2-mCherry) into NAc and terminated the doxycycline supply between day 0 and 21 after CCI surgery or CFA injections, such that mCherry-tagged Channel Rhodopsin 2 (ChR2) were preferentially expressed in NAc neurons with high expression of c-Fos, a molecular proxy of pain-induced neuronal activities (Figs. 1a and 3a). These c-fos-tTA mice were maintained using food or water containing doxycycline (40 mg per kg or per L) before CCI surgery or CFA injection. These mice express the tetracycline-transactivator (tTA), which is under the control of an activity-dependent c-fos promoter. In the absence of doxycycline (Dox), activation of the c-fos promoter leads to tTA expression, and tTA binding to the tetracycline-responsive element (TRE) induces the expression of an effect gene. However, the administration of Dox prevents the binding of tTA to TRE and silences the expression of the TRE-controlled transgene.

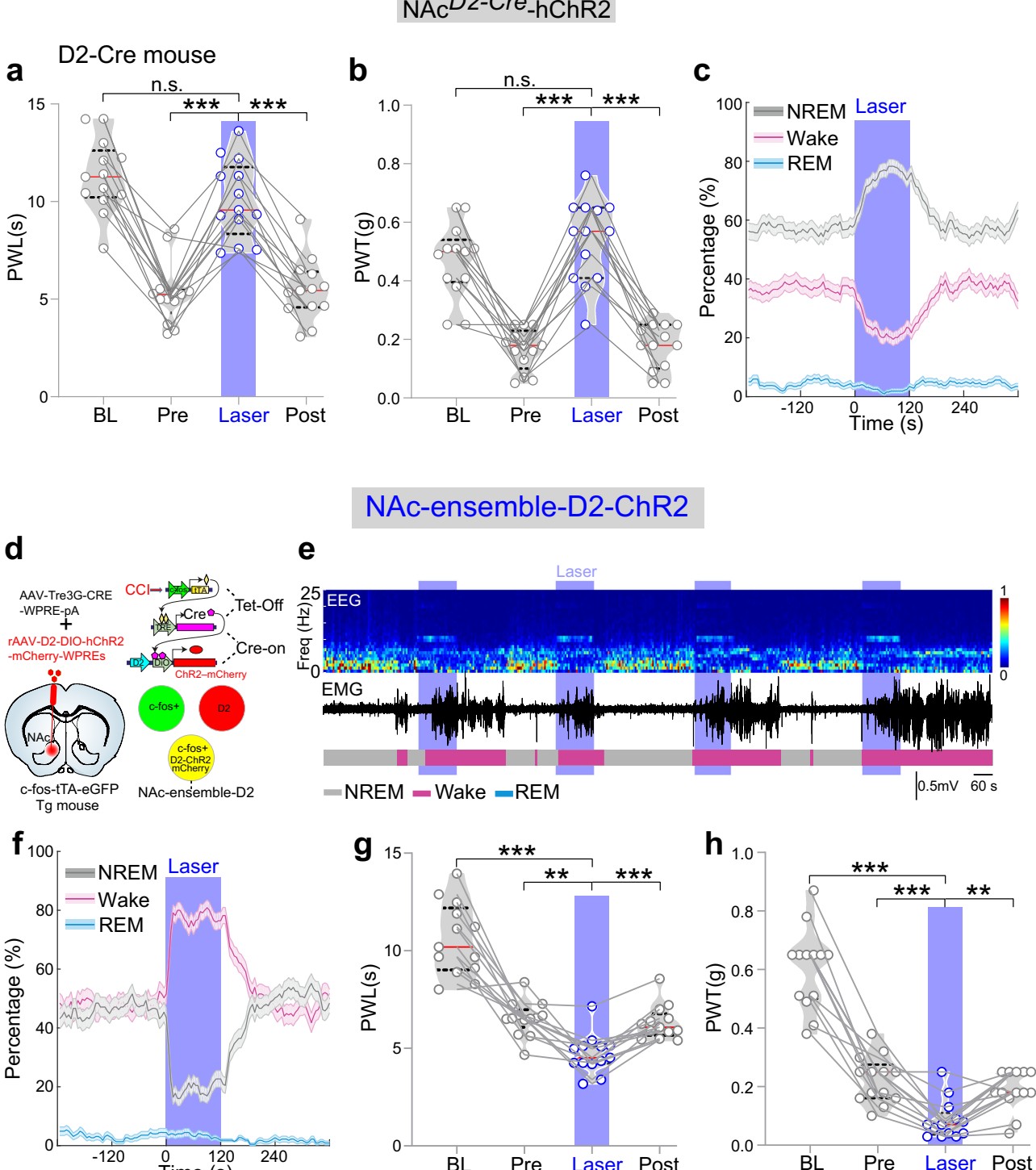

**Fig. 6 | The differences in regulating pain and wakefulness between activations of NAc-D2 and NAc-ensemble-D2 neurons. a, b** Optogenetic activation (473 nm, 10 Hz) of NAc D2 neurons significantly increases thermal PWL ($n = 13$ mice) and mechanical PWT ($n = 13$ mice) to the baseline levels in D2-cre mice. **c** Percentage of time in different brain states before, during, and after blue laser activation of NAc D2 neurons (473 nm, 10 Hz, 120 s; $n = 13$ mice). Note the significant increase in NREM sleep ($p < 0.0001$, bootstrap) and decrease in wakefulness ($p < 0.0001$) and REM ($p < 0.001$) sleep during laser stimulation. **d** The strategy for specifically labeling D2 neurons in NAc-ensemble neurons by combinating "Tet-Off" with "Cre-On". Brain figure was adapted from Allen Mouse Brain Atlas (http://atlas.brain-map.org/). **e** Representative EEG spectrogram (top), relative EMG trace (middle), and brain states (bottom) from a NAc-ensemble-D2-ChR2 mouse during baseline conditions and blue laser activation of NAc-ensemble-D2 neurons. Blue stripe indicates

laser stimulation (473 nm, 10 Hz, 120 s). **f** Percentage of time in different brain states before, during, and after blue laser activation of NAc-ensemble-D2 Neurons (473 nm, 10 Hz, 120 s). Note the dramatical decrease in NREM sleep ($p < 0.0001$, bootstrap), increase in wakefulness ($p < 0.0001$) no obvious change in REM sleep during laser stimulation ($p > 0.05$), $n = 13$ mice. **g, h** Optogenetic activation (473 nm, 10 Hz) of NAc-ensemble-D2 Neurons significantly decreases thermal PWL ($n = 13$ mice, $F_{(1.605, 19.26)} = 61.19$, $p < 0.0001$, BL vs. Laser, $p < 0.0001$, Pre vs. Laser, $p = 0.001$, Post vs. Laser, $p < 0.0001$) and mechanical PWT ($n = 13$ mice, $F_{(1.749, 20.99)} = 90.74$, $p < 0.0001$, BL vs. Laser, $p < 0.0001$, Pre vs. Laser, $p = 0.0003$, Post vs. Laser, $p = 0.0036$) in c-fos-tTA-Tg mice. Repeated measures one-way ANOVA test with Bonferroni's multiple comparisons test for (**a, b, g, h**). $*p < 0.05$, $**p < 0.01$, $***p < 0.001$, ns not significant.

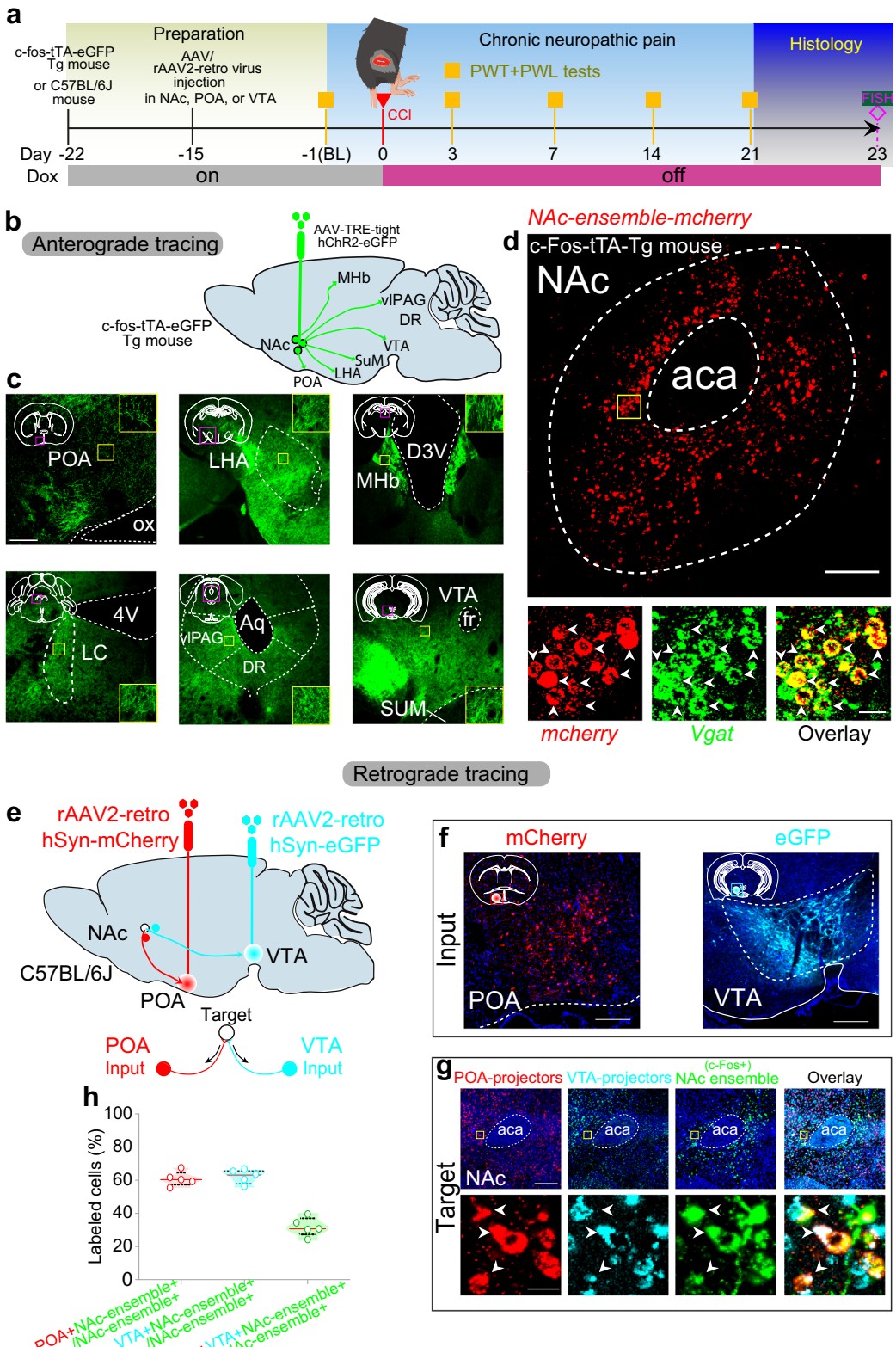

## Behavioral tests

**Paw withdraw threshold (PWT).** The 50% withdrawal threshold (50% PWT) was determined[80,81]. Briefly, mice were placed in polyethylene cages separately on an elevated metallic wire mesh platform. Before testing, mice were allowed to acclimate to the environment for 1 h. Mechanical allodynia of the hind paws was assessed using a series of calibrated von Frey filaments (USA, North Coast). Testing was initiated with the midrange filament of 0.16 g strength. Subsequent filaments were decided according to the up-down method. Each filament was applied to the mid-plantar surface of the hind paw until bending and maintained for 3 s. A positive response was considered when a mouse withdrew or licked its paw, and testing was conducted blind to experimental groups. Finally, 50% PWTs were calculated through the formula described previously[81].

**Fig. 7 | NAc ensemble neurons project to both POA and VTA. a** Experimental scheme showing AAV or retro-AAV injections and CCI surgery in c-fos-tTA transgenic or C57BL/6J mice. FISH was performed at day 23 after the CCI surgery. **b** Schematic drawing showing the axonal distributions of NAc ensemble in c-fos-tTA transgenic mice. NAc, nucleus accumbens; POA, preoptic area; LHA, lateral hypothalamus area; MHb, medial habenula; LC, locus coeruleus; vlPAG, ventrolateral periaqueductal gray; DR, dorsal raphe; VTA, ventral tegmental area; SuM, supramammillary region. **c** The dominant projections of NAc ensemble were represented in a series of coronal sections. Scale bar: 250 μm. ox, optic chiasm; 3V, 3rd ventricle; 4V, 4th ventricle; Aq, cerebral aqueduct; fr, fasciculus retroflexus. **d** Top: Expression of mCherry in NAc ensemble from CCI mice. Bottom: Fluorescence image (enlarged view of the region in the yellow box) showing double FISH of mCherry (red) and Vgat (green). Arrowheads indicate co-labeled neurons. Scale bar: top, 200 μm; bottom, 20 μm. **e** Schematic of viral injection for simultaneous retrograde tracing from POA and VTA in C57BL/6J mice. Brain figures in (**b** and **e**) were adapted from Allen Mouse Brain Atlas (2011 Allen Institute for Cell Science, Allen Mouse Brain Atlas, available at http://atlas.brain-map.org/). **f** Representative images showing retro-AAV injection sites in POA (red) and VTA (cyan). Scale bar, 300 μm. **g** Fluorescence images of NAc showing the retrograde labeled POA-projecting cells (red), VTA-projecting cells (cyan), and NAc ensemble expressing c-Fos. Scale bar, top, 200 μm; bottom, 20 μm. **h** Quantification of the percentage of labeled cells in NAc. Note that POA-projecting and VTA-projecting NAc ensemble are partially overlapping. $n = 5$ mice. POA+, POA-projecting cells; VTA+, VTA-projecting cells; NAc ensemble+, NAc ensemble expressing c-Fos.

**Paw withdraw latency (PWL).** PWLs were measured using the Hargreaves test[82] with an IITC plantar analgesia meter (IITC Life Science). In brief, mice were placed individually in polyethylene cages on a glass platform and allowed to acclimate to the testing apparatus for 1 h. A radiant heat source beneath the glass was used to stimulate the plantar surface of the hind paw. A positive response was considered when a mouse trembled, withdrew, or licked its paw, and the PWL was calculated as the average of three applications.

**Open field.** The general locomotor activity before and after activation of NAc ensemble neurons was measured using the open field. Each mouse was placed in an open field chamber (50 cm long × 50 cm wide × 30 cm high) for 10 min[83]. 24 h later, the same group of mice with optogenetic activation (blue laser, 473 nm, 10 Hz) or inactivation (yellow laser, 589 nm, 8 s on/2 s off) were repeated for open field test. Total moving distance was recorded using VisuTrack Animal behaviour analysis system (XinRuan Co. Ltd, Shanghai).

**Brain states recording**
For habituations, mice were allowed to habituate to the recording chamber and to attachment of the recording cables for a minimum of two days. For recordings, EEG and EMG electrodes were connected to recording headstages via flexible recording cables. EEG/EMG signals were amplified, filtered (0–500 Hz), digitized at 1500 Hz, and recorded using a NeuroLego amplifier (Jiangsu Brain Medical Technology Co. Ltd, Nanjing, China). Spectral analysis has been performed using the fast Fourier transform (FFT) in the EEG signals. The sleep-wake states were automatically classified by 5-s epochs as wake, NREM, or REM sleep with MATLAB according to published standard criteria[34,84]. Briefly, wakefulness was identified by the presence of desynchronized EEG and high EMG activity. NREM sleep was defined as synchronized EEG, high-amplitude, low-frequency (0.1–4 Hz) activity, and low EMG activity. REM sleep was characterized by regular theta rhythm (5–10 Hz) with low EMG activity. After habituation, C57BL/6J mice underwent 24 h undisturbed EEG and EMG recording starting at light onset in Supplementary Fig. 1d–g. The analyzers were blind to any information about the identity of the animal or laser stimulation timing.

**Optrode and tetrode recordings for extracellular single-unit activities in freely moving mice**
To identify NAc ensemble of freely moving mice, optrodes were prepared using a custom-made optrode mold[34,84]. Briefly, the optrodes consisted of an optic fiber with 200 μm diameter (Thorlabs, FT200-UMT) fused to 6 pairs of stereotrodes. The stereotrode was constructed by twisting two platinum-10%-iridium wires (35 μm in diameter, California Fine Wire, CA, USA) with an impedance of ~250 kΩ. Similarly, each tetrode consisted of four platinum-10%-iridium wires twisted together. The screw was mounted on both the optrode and tetrode so that it could be moved vertically using screw-driven. Single-unit activities were recorded while brain states were continuously

monitored (EEG and EMG). Unit signals were amplified, filtered between 0.3 Hz and 8 kHz, digitized at 25 kHz, and acquired by a NeuroLego amplifier (Jiangsu Brain Medical Technology Co. Ltd, Nanjing, China). Optrodes were advanced in 50 μm increments, up to 150 μm per day. After completing the recordings, an electrolytic lesion was made to mark the end of the electrode tract by passing a current (100 μA, 10 s) through two electrodes.

Spikes from these single neurons were isolated by cluster analysis using Offline Sorter (https://plexon.com/products/offline-sorter/). Spikes were sorted into clusters offline based on their waveform energy, peak amplitudes, and the first 3 principal components of the spike waveform on each stereotrode channel. Clusters were considered single units only when the following criteria were met: (1) refractory period (2 ms) violations were less than 0.2% of all spikes and (2) isolation distance, estimated as the distance from the center of the identified cluster to the nearest cluster based on the Mahalanobis distance, was more than 20. Only stable and well-isolated units were used for further analysis. Subsequent data analyses, such as comparison of average firing rates, firing rate histograms, and rastergrams were performed with Matlab (Mathworks, 2017) and NeuroExplorer software (v_5.0 & v_5.2).

Identification of NAc ensemble was performed following established methods[34,84]. Briefly, 10 Hz (duration 1 s/train) laser pulse trains (duration 1 ms/pulse) were delivered intermittently (interval 90 s). As shown in Fig. S2, ChR2-expressing neurons were identified if spikes were evoked by laser pulses with high reliability (>0.7 for all units in our sample), short first-spike latency (<3 ms for all units in our sample), low jitter (<3 ms for all units in our sample), and the waveforms of the laser-evoked and spontaneous spikes were highly similar (correlation coefficient > 0.95).

**Optogenetic manipulations**
Before optogenetic manipulations, mice received at least three 1-h habituations to the test environment. During optical stimulations, blue 473-nm or yellow 589-nm laser light was delivered by the laser (Shanghai Laser & Optics Century, China) through a 200 μm diameter optic fiber (Inper, Hangzhou, China), which was connected by a waveform generator (NeuroStim, Jiangsu Brain Medical Technology Co. Ltd, Nanjing, China). Specifically, for optogenetic activation of NAc ensemble and Non-ensemble in Figs. 2, 3 and Supplementary Fig. 5–6, blue laser pulses with a pulse width of 473 nm, power of ~5 mW, and frequencies of 5, 10, and 20 Hz (10 Hz for Figs. 2, 3, 5, 6, and 20 Hz for Supplementary Fig. 6) were delivered randomly from a uniform distribution between 4 and 10 min. Similarly, for optogenetic inactivation of NAc ensemble, D1 neurons, NAc ensemble→POA neurons, NAc ensemble→VTA neurons in sleep-wake states (Figs. 4d–f, 5i, 8f, g, and 8l, m) yellow laser pulses with a pulse width of 589 nm, power of ~10 mW, 8 s on/2 s off, and duration of 2 min were delivered randomly from a uniform distribution between 4 and 10 min. Furthermore, a bootstrap procedure was performed to evaluate whether a significant change in brain state was caused by laser application. In addition, we

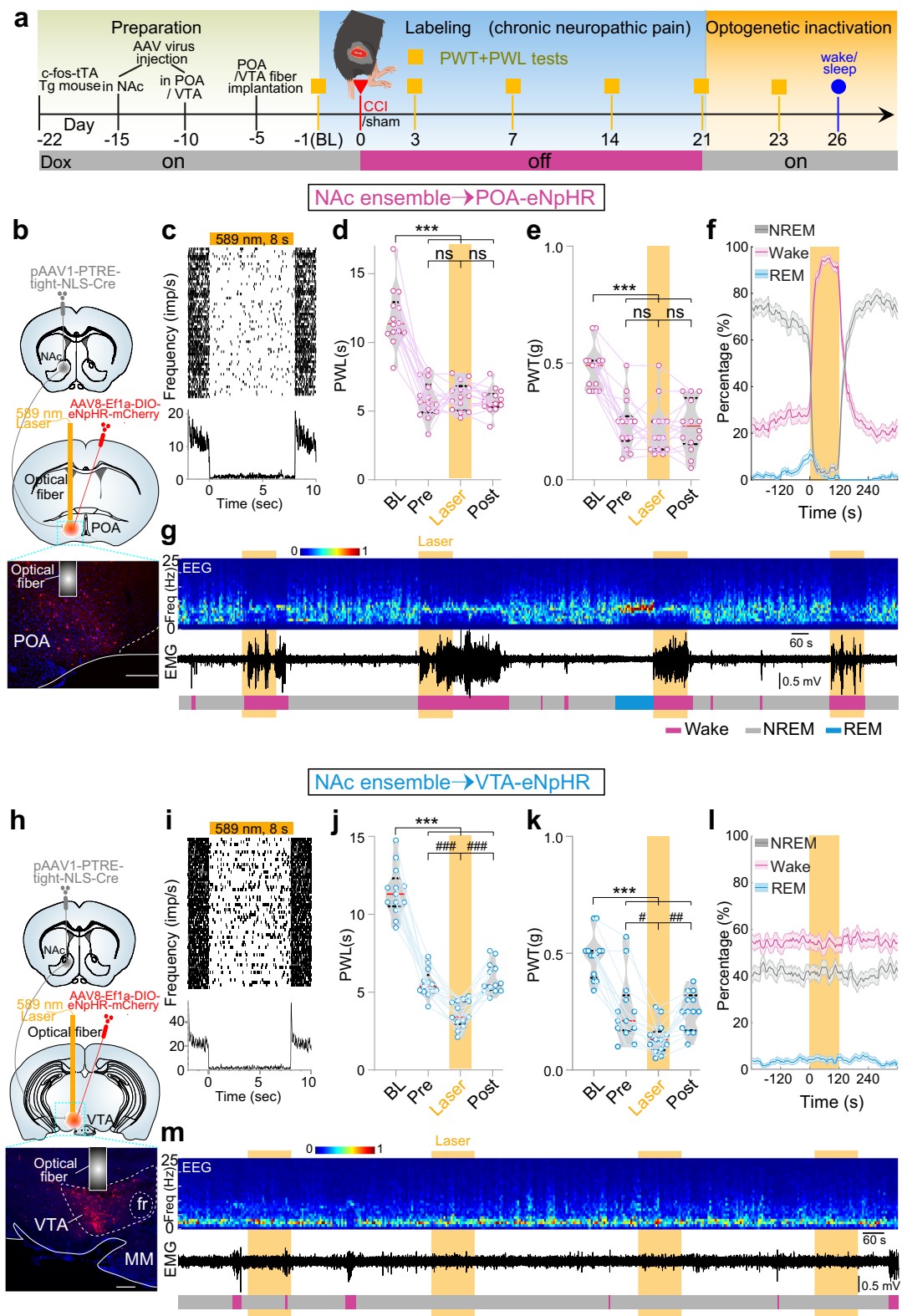

calculated the difference between the mean probabilities during laser application and the preceding period of identical duration using each bootstrap iteration and computed the 95% confidence interval of this difference distribution. For silencing NAc ensemble, D1 neurons, NAc ensemble→POA neurons, NAc ensemble→VTA neurons in mediating nociception (Figs. 4b, c, 5g, h, 8d, e, and 8j, k), we applied yellow laser pulses (0.1 Hz, duration 10 s, pulse width 8 s, delay 2 s, -10 mW).

## Sleep deprivation protocol

C57/BL6J mice and c-fos-tTA mice were kept awake by gentle stimulations (such as an enriched, novel environment, novel objects, or tapping/shaking lightly the cages). Mice were never touched directly to avoid stress. Sleep deprivation was performed at ZT 0 and persisted for 6 h during the daytime. To make sure that each mouse was fully awake, we closely monitored EEG and EMG signals during this period.

**Fig. 8 | Inhibition of NAc-ensemble-innervated POA or VTA neurons.**
**a** Experimental scheme. **b** Schematic showing the injection of pAAV1-PTRE-tight-NLS-Cre into NAc and the injection of AAV8-Ef1a-DIO-eNpHR-mCherry into POA, respectively. NAc ensemble→POA projecting Cre-dependent enhanced natronomonas pharaonic halorhodopsin (eNpHR) validation 4 weeks after injection.
**c** Raster plot (top) and PSTH (bottom) of a NAc-ensemble-innervated POA neuron showing the firing activity before, during, and after 8 s optogenetic silencing.
**d, e** Optogenetic inactivation of NAc-ensemble-innervated POA neurons had no effect on the thermal PWL ($n = 14$ mice) and mechanical PWT ($n = 14$ mice).
**f** Optogenetic inactivations of NAc-ensemble-innervated POA neurons (589 nm, yellow laser, 8 s on/2 s off, 120 s) significantly decrease in NREM sleep ($n = 11$ mice, $p < 0.0001$, bootstrap), increase in wakefulness ($p < 0.0001$), and do not affect REM sleep ($p = 0.363$). Shading represents ± SEM. **g** Similar to (Fig. 4f), but for

inactivation of NAc ensemble-innervated POA neurons. **h** Similar to (**b**), but with injection AAV into VTA. Brain figures in (**b** and **h**) were adapted from Allen Mouse Brain Atlas (2011 Allen Institute for Cell Science, Allen Mouse Brain Atlas, available at http://atlas.brain-map.org/). **i** Similar to (**c**), but for a NAc ensemble-innervated VTA neurons. **j, k** Optogenetic inactivations of NAc ensemble-innervated VTA neurons significantly decrease the thermal PWL ($n = 13$ mice) and mechanical PWT ($n = 13$ mice). **l** Optogenetic inactivations of NAc ensemble-innervated VTA neurons (589 nm, yellow laser, 8 s on/2 s off, 120 s) do not affect NREM sleep ($n = 13$ mice, $p = 0.490$, bootstrap), wakefulness ($p = 0.132$), and REM sleep ($p = 0.140$). Shading represents ± SEM. **m** Similar to (Fig. 4f), but for inactivation of NAc ensemble-innervated VTA neurons. Repeated measures one-way ANOVA test with Bonferroni's multiple comparisons test for (**d, e, j, k**). ***$p < 0.001$, #$p < 0.05$, ##$p < 0.01$, ###$p < 0.001$, ns not significant.

---

When EEG and EMG signals indicated NREM sleep, we applied these gentle stimulations to mice. Mice were not disturbed when they were spontaneously awake.

## Quantification and statistical analysis

**Brain state transition probabilities.** The transition probability was computed at a 60 s time bin[34]. Briefly, all the trials (n) in which the animal was in state X (X may be wakefulness, NREM, or REM) in the preceding time bin (i-1) were chosen. Among these n trials, the subset of trials (m) in which the animal transitioned into state Y (Y may be wakefulness, NREM, or REM) in the current time bin (i) were then counted. Lastly, we calculated the transition probability at time bin (i) for X → Y as m/n. In addition, the 95% confidence intervals (CI) for brain state transition probabilities were calculated using a bootstrap procedure. Specifically, for an experimental group of n mice, with mouse i comprising $m_i$ trials, we first repeatedly resampled the data by randomly drawing for each mouse $m_i$ trials (random sampling with replacement). Next, the mean probabilities for each brain state transition were recalculated across the n mice for each of the 10,000 iterations. We then extracted the lower and upper confidence intervals from the distribution of the resampled mean values. After that, we also used bootstrap to test whether a given brain state is significantly modulated by laser stimulation. Finally, the differences between the mean probabilities during laser stimulation and the preceding period of identical duration were calculated for each bootstrap iteration.

## Statistical analysis and reproducibility

For optogenetic and behavioral experiments, mice were randomly assigned to control/sham and experimental groups. All summarized data in violin plots were presented as median (red line) with 25th and 75th percentile (dash line). The error bars in time courses and total distance traveled were SEM (Supplementary Figs. 1e1–g1 and 7g–h). The error bars in brain state transition probabilities were 95% confidence intervals (Supplementary Figs. 4c, d, 5e, and 10d, e). Two-way ANOVA was used to assess significant interactions of group vs time (Figs. 1i, 1k, and 3b; Supplementary Figs. 1b, c, e₂-g₂, 6b, and 12d). Briefly, we first calculated the effect size and defined it as the difference between two groups (sham and CCI) or among three groups (sham, identified, and unidentified) and each time point. We then performed two-way ANOVA (group × time). After confirming significant the main effect of the group, Bonferroni post hoc test was used to identify each time point at which the difference between groups was significant. One-way ANOVA test was applied for comparing more than two different groups or more than two different time points in the same group, followed by multiple comparisons with Bonferroni's correction (Fig. 1l). Repeated one-way measures ANOVA test was used for paired data at different time points in the same group (Figs. 2c, d, I, j, 3c, 4b, c, 5d, e, g, h, 6a, b, g, h, 8d, e, j, k; Supplementary Figs. 5a, b, 6b, 8b, c, 9b, c, 10a, b, 13a, b, d, e, 14c, d, and 14f, g). Paired t test was used to compare the same group of mice before and after laser stimulations (Supplementary Fig. 7g, h).

Mann–Whitney test was performed to compare the differences between two independent groups when the sample distributions were not normally distributed and the sample sizes were small (Supplementary Fig. 2b). In graphs, statistical significance is indicated by asterisks. */# for $p < 0.05$, **/## for $p < 0.01$, and ***/### for $p < 0.001$. Statistical analyses were performed using GraphPad Prism 8.0., SigmaPlot 14.0, Matlab 2017, SPSS V22. The images in Figs. 2b, h; 7c, d, f, g are representative ones among all from mice we collected data in individual experiments. Each experiment has been repeated at least three times in the manuscript with consistent results. Source data in all figures are provided in a Source data file submitted with this paper.

## Reporting summary

Further information on research design is available in the Nature Portfolio Reporting Summary linked to this article.

## Data availability

The complete data set generated in this study is described and provided in this document, in the Supplementary Information and the Source data file. Because further raw data are huge and presented in highly diverse nature and formats, these raw data are available from the corresponding author upon request. Source data are provided as a Source data file. Source data are provided with this paper.

## Code availability

Custom codes are available on GitHub (https://github.com/brain-state-analysis/-Sun-2023_Nature-Communications.git).

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

## Acknowledgements

We thank Prof. Werner Kilb from Mainz University in Germany for his comments on the manuscript, Cui Yin, Junxia Yang, and Lingyun Hao from Jiangsu Province Key Laboratory of Anesthesiology, Zhou Xu and Zhiwei Liu from Public Experimental Research Center of Xuzhou Medical University for their expert assistance. This study was supported by the National Natural Science Foundation of China (82101309 to H.S., 31970944 to S.A., 82130033, 81720108013, and 81230025 to J.-L.C.), the Sci-Tech Innovation 2030 (2021ZD0203100 to J.-L.C.), the Foundation for Jiangsu Province Specially Appointed Professors (S.A.), Scientific Research Foundation for Excellent Talents of Xuzhou Medical University (TD202203, D2019036 to S.A.), the Foundation for high level innovative and entrepreneurial talents in Jiangsu Province (to H.S.), Scientific Research Foundation for Excellent Talents of the Affiliated Hospital of Xuzhou Medical University (to H.S.), the Key Project of Nature Science Foundation of Jiangsu Education Department (11KJA320001 to J.-L.C.).

## Author contributions

H.S., S.A., and J.L.C. designed research; H.S., Z.L., Z.Q., Y.S., and S.A. performed research; H.S., Z.L., Q.G., S.A., and J.L.C. analyzed data; and S.A., H.S., S.W.H., H.L.D., and J.L.C. wrote the paper.

## Competing interests

The authors declare no competing interests.
