## [Peer Review File · Nature Communications]

A common neuronal ensemble in nucleus accumbens regulates pain-like behaviour and sleepREVIEWER COMMENTS

Reviewer #1 (Remarks to the Author):

The study by Sun et al. used two chronic pain models in mice to demonstrate that a dual-functional NAc ensemble is assembled in chronic pain that regulates both pain and sleep. They further characterized the segregation of the two functions via distinct but overlapping downstream projections. The study is conceptually provocative and has broad implications for understanding the functional modules and interactions between pain and sleep regulations. The experimental design is comprehensive with multiple layers of controls, and the results provide strong support to the main conclusions. The manuscript is carefully written, and the narrative is clear for the most part. I have a few minor concerns as follows:

1. It is not mentioned or discussed when c-Fos is expressed between day 0 and 21 after CCI surgery or after CFA injections – given that c-Fos is often transiently induced, in contrast to the prolonged Tet-off periods. It will be helpful to cite previous work (if any) that documented this.
2. One alternative interpretation for the decreased PWL could be increased alertness (wakefulness), and not necessarily increased nociception. However, mimicking NAc-POA ensemble activities increased wakefulness but did not reduce PWL (Fig. 7D-F), suggesting that the authors' interpretations hold. This can be discussed.
3. Fig. 1M: Wake EEG in the 3 panels shows different characteristics (1: high theta low delta; 2: low theta, low delta; 3: low theta, high delta). Not sure if they are all representative.
4. Fig. 3F: Not clear if the two example images may be directly comparable -- judging from the appearance of the "aca" area the exposure times may be very different.
5. Fig. 6: It is not clear which set of experiments used which one of the two mice lines.
6. Fig. 6H: Seems a missed opportunity to report on % of POA+NAc-ensemble+/NAc-ensemble+; and % of VTA+NAc-ensemble+/NAc-ensemble+. These measures will complement the result on %overlapping population (POA+VTA+NAc-ensemble+/NAc-ensemble+), which may help us better estimate the potential interactions between sleep vs pain regulatory modules.
7. Fig. 7B,H: It is desirable to show a control condition – that mCherry is not expressed in the absence of Cre.

Reviewer #2 (Remarks to the Author):

Sun et al. used cutting-edge methodological approaches, an impressive array of experiments, and seemingly perfect results to answer a key question in the field of the co-morbidity of chronic pain and sleep disturbance. Their conclusion of the study is NAc D1-positive neurons control both chronic pain and pain-induced sleep disorders via parallel projections, which was supported by following 3 results: 1. NAc D1R neurons increased firing in both wakefulness and pain. 2. Manipulation of the pain 'ensembles' causally alters pain and sleep, 3. NAc D1R neurons to POA and VTA projection regulated wakefulness and pain independently. The experiments presented in this study seem well performed, Figures are

nicely illustrated and the manuscript is clearly written.

However, the shortage of this study is neither of NAcD1R in regulating arousal nor in regulating pain is novel. Amble studies demonstrated that NAc is a key node in controlling wakefulness with motivation and sleep without motivation. On the other hand, NAc is also involved in chronic pain regulation and pain-related depression. The authors' findings may not be beyond these reports, but these findings still provide novel information to the field.

The key question for the current study is the conclusion that the NAc ensemble encodes chronic pain and controls sleep. However, current results are not sufficient to support the role of 'encoding', in other words, whether chronic pain-induced sleep disorders through activation of NAc D1R neurons can not rule out from the data so far, which reduces the significance of conclusions. My comments listed below may help authors to improve their study.

Major concerns

1. The author showed that chronic pain activated ~80% NAc D1R neurons, and activation of these neurons induced wakefulness. However, previous studies showed NAc D1R neuron activation induced long-lasting wakefulness, which means that optogenetic activation of D1R 'ensembles' in the current study may be over the range of pain itself. The authors also report that activation of non-ensembles did not alter sleep, which could be due to the cell number being relatively low, considering 80% of D1R neurons are ensembles or they activated different subpopulation NAc neurons. This is worth discussion. What is harder to understand is nonspecific activating ChR2-expressing NAc neurons did not alter sleep-wake behaviors, this is challenging the sleep field that NAc is a sleep-wake regulation center because several studies with gain- or loss-of-function methods demonstrated that NAc plays a dominate role in regulating arousal in multi-conditions and several central wake-promoting regions, such as the VTA and PVT, regulate wakefulness through NAc.
2. NAc is also involved in chronic pain regulation. A very recent study (Sato, 2022) indicates that optogenetic activation of NAc D1R neurons reversed the lowered pain thresholds due to neuropathic pain, the inconsistent findings with this study might be due to different animal models and manipulation of different populations of D1 neurons. Nevertheless, NAc D1 neurons regulate chronic pain. The author should consider how to interpret their findings are different from previous studies, instead of saying 'Our results show that some of these sleep-regulating NAc neurons also regulate pain experience'. If the population of cells involved in the regulation of behavior also regulates the send relevant behavior, the significance of this study is discounted.
3. When the authors are talking about the NAc D1 neurons, I strongly feel that they are avoiding exposing that NAc D2 neurons are also involved in sleep and pain regulation. The authors reported that only ~20% of D2 neurons are labeled by chronic pain, which is not well consistent with a previous study that chronic pain dramatically increased the AMPA/NMDA ratio in NAc D2 neurons (Ren, 2016), indicating an increase of D2 neuronal activity. Furthermore, 20% of neurons can play an important role in regulating both sleep and pain because activation of D2 neurons increases sleep (Oishi, 2017). Chronic pain activates D2 neurons and D2-neuron activation increases sleep are NOT in the context of chronic pain-induced sleep disorders. The author should carefully address this gap but not avoid talking about these issues.
4. The authors tried to explain NAc D1R neurons control chronic pain via VTA projection, but emphasize that this pathway is not involved in regulating wakefulness. This is also out of my understanding. Do the

pain-labeled D1 neurons significantly different from other NAc D1 neurons because the previous study showed that NAc D1-VTA projection promotes wakefulness (Luo, 2018). If yes, it is necessary to show the difference. In addition, whether optogenetic activation of NAc-VTA projection increases the activity of VTA dopaminergic and glutamatergic neurons? It's better to check the activity of these neurons after optogenetic activation to further explain the downstream circuit mechanisms.

5. Same experiments in Fig.5 are suggested to perform in NAc D2 neurons.

Minors

1. Are there any D2 neurons labeled in Fig1B also activated wakefulness? Maybe the author can present D1 and D2 neurons activity across sleep-wake transitions in a supplementary figure.
2. Figure 1F & 2D, a significant 10 Hz signal in EEG heatmap. Is it from opto-stimulation? However Figure 1L did not show it. Noise elimination is needed to increase the reliability of results.
3. In Figure S3, the firing rate of NAc D1 neurons was not altered in sleep-wake transitions in naïve mice, do the author think these neurons are not involved in the regulation of wakefulness? How to understand the increased calcium activity of NAc neurons in wakefulness from the previous study?
4. Figure 6H, why these VTA and POA-projecting cells are normalized to total VTA and POA cells, but not NAc cells?
5. Inhibition experiments in Figure 7 are indirect evidence. Whether the stimulation of NAc to POA and VTA projections will get similar results?

Reviewer #3 (Remarks to the Author):

This is an elegant study that significantly advances our knowledge about the interplay between sleep and pain. I will summarize the main data to let the authors know what I believe they should consider before publication.

Sun and cols. used c-fos-tTA transgenic mice to express several AAV-carried effect genes under the c-fos promoter, so that the transgenes were preferentially expressed in neurons with high expression of c-Fos in the target region. The authors focused on the Nucleus accumbens (NAc) an interface region in the control of movement, pain, and sleep-wake cycle. The expression of the controlled transgenes was released (by terminating doxycycline supply) between day 0 and 21 after CCI (chronic constriction of the sciatic nerve) or CFA (Complete Freund's adjuvant) injections. So, NAc neurons activated in these chronic pain models were targeted.

These neurons showed naturally increased activity during nociceptive tests and wakefulness (with decreased activity during non-REM sleep (NREM sleep)) (Fig 1). The optogenetic stimulation of these neurons further decreases mechanical paw withdrawal threshold (PWT) and thermal latency (PWL); and increases wakefulness (decreasing NREM sleep) in CCI or CFA animals (Fig 2 and 3). Complementary, their optogenetic inactivation induces the opposite effects (Fig 4). The majority (80%) of CCI target neurons express dopamine D1 receptors and optogenetic stimulation of these D1 neurons (using D1-Cre

mice) decreased both the PWT and PWL and increased the percentage of wakefulness (decreasing sleep); while their optogenetic inactivation increases the PWT and PWL and increases sleep (Fig 5). Anterograde tracing of NAc target neurons identified several downstream brain regions that are known to regulate sleep/wakefulness and/or nociception and a double-labeling assay suggest that these neurons are GABAergic principal medium spiny neurons. The authors focused in VTA (ventral tegmental area) and POA (preoptic area) and used a retrogradely label strategy to demonstrate that a portion (30%) of NAc target neurons divergently projected to both POA and VTA (fig 06). Finally, the authors used a dual-viral system to optogenetically inactivate the VTA or POA neurons innervated by the NAc targeted neurons. Inactivation of POA neurons decreases sleep (with no effect on nociceptive tests), while inactivation of VTA neurons decreases PWT and PWL (with no effect on sleep) (fig 07). Based on this broad set of data, the authors concluded that these NAc-targeted neurons form "... a common NAc ensemble that encodes chronic pain and controls sleep"

The bidirectional relationship between negative changes in sleep and pain has been the focus of several studies in the last two decades and the NAc is emerging as a key candidate to mediate the effect of decreased sleep on pain processing. However, the underlying mechanisms are poorly understood, and this is the first study to identify a neural population in the NAc that is apparently involved in the control of pain and sleep/wake cycle. These data will impact the literature on the field and are at the edge of knowledge. I have one main concern about the experiments and some suggestions to improve the manuscript reading and data presentation

Major

How are the authors sure that the NAc targeted neurons does not simply control the movement of paw withdrawal after mechanical or thermal stimulation in CCI or CFA animals? This is important not only in view of the role of NAc (especially D1 neurons) in initiating movement, but also because paw withdrawal was consistently evoked throughout the experiments with no apparent need, since these behavioral data (days 3 to 21) were not shown. Indeed, it is unfortunate that the authors did not perform a single experiment to assess motor function (another than PW) upon targeted neurons manipulation. I believe that the absence of PW changes in sham animals does not solve the problem because they are not under neuropathic (CCI) or inflammatory (CFA) sensitization. Some kind of motor function monitoring would also be welcome during optogenetic stimulation that induces the transition from sleep to wakefulness. This reviewer would like to see at least one nociceptive experiment performed without motor bias. Conditioned place preference (CPP) under target neurons inhibition would be a great alternative and it is feasible in CCI model.

The experimental strategy is based on targeting NAc neurons that express c-Fos after CCI or CFA, but there is no comparison of the expression of c-Fos (or associated fluorescent proteins) between CCI/CFA and sham animals. Such a comparison would be welcome to assess the effect of CCI/CFA on NAc activity.

Minor

Abstract: The chronic pain models used should be mentioned in the abstract.

Results:

Please use the same scale on the Y-axis for similar experiments throughout the presentation of results. The lack of this standardization makes it difficult to compare data between different figures. Baseline values for PWT / L should be plotted in figures, as it was done in figure 4 B and C. Fig S1 E1-G1 there is no red plot or line

Please indicate that FISH means (Fluorescent in situ Hybridization) in the first figure legend

Methods:

Why were the NAc injections unilateral? Is there any laterality between the injected side (NAc) and the side subjected to CCI/CFA (paw)?

There is no mention to mice used in experiments of figure 5 (Cre) and 6 (black)

Discussion

“Our results show that neurons in the dual-functioning ensemble in CCI mice exhibited several cellular dynamics distinct from non-ensemble neurons (Figures 2B and 2H)” Figures 2 B and H are not related to these findings

Point-by-point response to reviewers' comments

REVIEWER COMMENTS:

Reviewer #1 (Remarks to the Author):

The study by Sun et al. used two chronic pain models in mice to demonstrate that a dual functional NAc ensemble is assembled in chronic pain that regulates both pain and sleep. They further characterized the segregation of the two functions via distinct but overlapping downstream projections. The study is conceptually provocative and has broad implications for understanding the functional modules and interactions between pain and sleep regulations. The experimental design is comprehensive with multiple layers of controls, and the results provide strong support to the main conclusions. The manuscript is carefully written, and the narrative is clear for the most part. I have a few minor concerns as follows:

Response : We appreciate the reviewer for your positive comments on our work.

Comment 1: *It is not mentioned or discussed when c-Fos is expressed between day 0 and 21 after CCI surgery or after CFA injections – given that c-Fos is often transiently induced, in contrast to the prolonged Tet-off periods. It will be helpful to cite previous work (if any) that documented this.*

Response 1: We appreciate the reviewer's insight. Recently, Jiang et al also prolonged Tet-off periods to label itch- and pain-specific neurons (Jiang et al., 2022). The c-Fos has been used as a neural marker of pain over several decades since Hunt et al (Hunt et al., 1987). CCI model as a model for chronic neuropathic pain, CCI mice exhibited persistent pain hypersensitivity at least for 3 weeks. During this period, more pain-transmitting neurons become active, react more intensely to stimuli, and grow more connections to second-order neurons (Voscopoulos and Lema, 2010). These neurons would express c-Fos repeatedly and alternately during constant pain, once the c-Fos was induced, it would be labeled by Tet-off system. We cite the previous work in the introduction on Page 4, line 4.

Comment 2: *One alternative interpretation for the decreased PWL could be increased alertness (wakefulness), and not necessarily increased nociception. However, mimicking NAc-POA ensemble activities increased wakefulness but did not reduce PWL (Fig. 7D-F), suggesting that the authors' interpretations hold. This can be discussed.*

Response 2: We thank reviewer for the kind suggestion. We add “All these results also suggested that decreased PWL and PWT (Figs. 2, 3, 5, 7) were not resulting from increased alertness.” in the discussion on Page 20, line 3-4.

Comment 3: *Fig. 1M: Wake EEG in the 3 panels shows different characteristics (1: high theta low delta; 2: low theta, low delta; 3: low theta, high delta). Not sure if they are all representative.*

Response 3: We thank reviewer for the insightful question. The different wake-EEGs during different brain state transitions were consistent with previous studies. Yu et al. investigated the activities of VTA neurons at the boundaries of brain state transitions (Yu et al., 2019). Interestingly, the wake-EEG exhibited high theta-power and low delta-power during NREM to Wake transitions. Wake-EEG showed high delta-power and low theta-power during REM to Wake transitions. Wake-EEG displayed medium delta-power and theta-power during Wake to NREM transitions. Furthermore, the wakefulness of mouse has been proposed to be divided into three sub-states with different EEG and motor activity: locomotion, nonlocomotor movement, and quiet wakefulness (Liu et al., 2020a). Locomotion was characterized by high theta-power, low delta-power in EEG, and high EMG power. The quiet wakefulness was associated with low theta-power, high delta-power in EEG, and low EMG power. Some nonlocomotor movement was characterized by low theta-power, delta-power in EEG, and medium EMG power. It would be interesting to investigate how do NAc-ensemble neurons fire during different sub-states of wakefulness in future study.

Comment 4: *Fig. 3F: Not clear if the two example images may be directly comparable – judging from the appearance of the “aca” area the exposure times may be very different.*

Response 4: We thank the reviewer's insight. We change new image for sham group in new Figure 3f. The two images in new Fig. 3f have the same exposure time, brightness, and contrast.

Comment 5: Fig. 6: It is not clear which set of experiments used which one of the two mice lines.

Response 5: We apologize for not being clear before. We clarify the mice used in new Fig 6b,6d,6e, and legend correspondingly.

Comment 6: Fig. 6H: Seems a missed opportunity to report on % of POA+Nac-ensemble+/Nac-ensemble+; and % of VTA+Nac-ensemble+/Nac-ensemble+. These measures will complement the result on % overlapping population (POA+VTA+Nac-ensemble+/Nac-ensemble+), which may help us better estimate the potential interactions between sleep vs pain regulatory modules.

Response 6: We appreciate the helpful suggestion from reviewer. To follow the reviewer's request, we have reanalyzed data and normalized the VTA and POA-projecting NAc ensemble neurons to all NAc ensemble neurons (POA+Nac-ensemble+ / Nac-ensemble+ and VTA+Nac-ensemble+ / Nac-ensemble+) in new Figure 6h. **Please also see our responses to minor comment 4 from reviewer #2.**

Comment 7: Fig. 7B,H: It is desirable to show a control condition – that mCherry is not expressed in the absence of Cre.

Response 7: We thank reviewer for the kind suggestion. We perform control experiment to inject AAV8-Ef1a-DIO-eNpHR-mCherry only into POA and VTA respectively, without injection of pAAV1-PTRE-tight-NLS-Cre into NAc. We find no mCherry expression in POA and VTA in the absence of Cre (**Figure R1**). The control condition is illustrated in new Supplementary Fig. S10a-b, and the new results are described on Page 15, line 6-9.

Figure R1. No mCherry expression in POA and VTA in the absence of Cre. *a* Schematic showing the injection of AAV8-Ef1a-DIO-eNpHR-mCherry only into POA of c-fos-tTA transgenic mice two weeks before CCI surgery. Right: a representative segment showing the overlap between DAPI and red-fluorescence in POA. Scale bar: 200 μ m. Note that no mCherry was expressed in POA without injection of pAAV1-PTRE-tight-NLS-Cre into NAc. *b* Similar to panel *a*, but for the injection into VTA in CCI mice.

Reviewer #2 (Remarks to the Author):

Sun et al. used cutting-edge methodological approaches, an impressive array of experiments, and seemingly perfect results to answer a key question in the field of the comorbidity of chronic pain and sleep disturbance. Their conclusion of the study is NAc DI-positive neurons control both chronic pain and pain-induced sleep disorders via parallel projections, which was supported by following 3 results: 1. NAc DIR neurons increased firing in both wakefulness and pain. 2. Manipulation of the pain 'ensembles' causally alters pain and sleep, 3. NAc DIR neurons to POA and VTA projection regulated wakefulness and pain independently. The experiments presented in this study seem well performed, Figures are nicely illustrated and the manuscript is clearly written. However, the shortage of this study is neither of NAcDIR in regulating arousal nor in regulating pain is novel. Amble studies demonstrated that NAc is a key node in controlling wakefulness with motivation and sleep without motivation. On the other hand, NAc is also involved in chronic pain regulation and pain-related depression. The authors' findings may not be beyond these reports, but these findings still provide novel information to the field. The key question for the current study is the conclusion that the NAc ensemble encodes chronic pain and controls sleep. However, current results are not sufficient to support the role of 'encoding', in other words, whether chronic pain-induced sleep disorders through activation of NAc DIR neurons can not rule out from the data so far, which reduces the significance of conclusions. My comments listed below may help authors to improve their study.

Major concerns

Comment 1: *The author showed that chronic pain activated ~80% NAc DIR neurons, and activation of these neurons induced wakefulness. However, previous studies showed NAc DIR neuron activation induced long-lasting wakefulness, which means that optogenetic activation of DIR 'ensembles' in the current study may be over the range of pain itself. The authors also report that activation of non-ensembles did not alter sleep, which could be due to the cell number being relatively low, considering 80% of DIR neurons are ensembles or they activated different subpopulation NAc neurons. This is worth discussion. What is harder to understand is nonspecific activating Chr2-expressing NAc neurons did not alter sleep-wake behaviors, this is challenging the sleep field that*

NAc is a sleep-wake regulation center because several studies with gain- or loss-of-function methods demonstrated that NAc plays a dominant role in regulating arousal in multi-conditions and several central wake-promoting regions, such as the VTA and PVT, regulate wakefulness through NAc.

Response 1: We appreciate the reviewer's insight. Generally, the NAc is composed of 90% GABAergic medium spiny neurons (MSNs) and some interneurons (Ins). Some of NAc-neurons express dopamine D1Rs, D2Rs, adenosine A2A receptors, oxytocin receptors, or choline acetyltransferase (Lazarus et al., 2011; Luo et al., 2018; Oishi et al., 2017; Tellez et al., 2012; Williams et al., 2020). Recently, Chen and his colleagues further classified the D1 and D2 MSNs into 30 D1 and 27 D2 subtypes by using single-cell RNA sequencing (Chen et al., 2021). They identified 7 subtypes neurons from NAc interneurons. These neuron subtypes may contribute to functional heterogeneity of the NAc. Besides sleep-wakefulness and nociception regulation, the NAc also plays important roles in higher brain functions involving reward, sensitization, addiction, feeding, social, and depression-like behaviors (Francis et al., 2015; Kai et al., 2015; O'Connor et al., 2015; Ren et al., 2016; Smith et al., 2013; Tellez et al., 2012; Williams et al., 2020; Zhou et al., 2019). In our study, we found that the majority of CCI-induced NAc ensemble expressed dopamine D1 receptor ($80.73 \pm 1.72\%$), and the minority of CCI-induced NAc ensemble expressed dopamine D2 receptor ($18.75 \pm 1.20\%$). NAc non-ensemble neurons may be composed of D1R, D2R, oxytocin, and adenosine A receptors neurons. These neurons may be sleep-promoting, wake-promoting, or don't have correlation with sleep-wakefulness regulation. Consistently, the non-specific neurons in NAc from sham mice and unidentified NAc neurons exhibited a much higher degree of functional diversity, including many units that were more active during wake, NREM or REM sleep, respectively (Figure 1k-l). Some non-specific neurons were neither wake-active nor sleep-active neurons. Activating all of them may have opposite or no effects on sleep and wakefulness, and may not alter sleep-wakefulness behaviors.

In sham mice, we used the same tTA-TRE (Tet-Off)-based viral system in c-Fos-tTA mice

as CCI mice (Figure 1a). And we terminated the doxycycline supply between day 0 and 21 after sham surgery, such that mCherry-tagged Channel Rhodopsin 2 (ChR2) were preferentially expressed in NAc neurons with high expression of c-Fos. The c-Fos expression level in NAc following sham surgery was significantly lower than that following CCI surgery (**Figure R10, please also see our responses to major comment 1b from reviewer #3**). During 3 weeks after sham surgery, NAc neurons may be activated and tagged with ChR2 by feeding, social, locomotion, or sleep-wake behaviors (Luo et al., 2018; O'Connor et al., 2015; Oishi et al., 2017; Williams et al., 2020; Zhu et al., 2016). Consistently, in our optrode recording, some tagged neurons in sham mice were neither wake-active nor sleep-active neurons (Figure 1k-l). Moreover, there was no significant difference in the mean firing rate of these tagged between wakefulness, NREM sleep, and REM sleep (Figure 1k). These identified neurons may be related with feeding, social, or locomotion behaviors. The mean firing rate of these neurons was not altered in sleep-wake transitions (Figure S3a). Reactivating these neurons may not regulate sleep-wakefulness states at all, but modulate other behaviors. These have been discussed in the Discussion section on Page 18, line 5-22 and Page 19, line 1-2. **Please also see our responses to minor comment 3.**

***Comment 2:** NAc is also involved in chronic pain regulation. A very recent study (Sato, 2022) indicates that optogenetic activation of NAc D1R neurons reversed the lowered pain thresholds due to neuropathic pain, the inconsistent findings with this study might be due to different animal models and manipulation of different populations of D1 neurons. Nevertheless, NAc D1 neurons regulate chronic pain. The author should consider how to interpret their findings are different from previous studies, instead of saying 'Our results show that some of these sleep-regulating NAc neurons also regulate pain experience'. If the population of cells involved in the regulation of behavior also regulates the send relevant behavior, the significance of this study is discounted.*

Response 2: We thank the reviewer for pointing out this issue. Sato et al did laser stimulations at 30Hz of NAc D1R neurons to reverse the lowered pain thresholds. In our study, application of blue laser at 10Hz to NAc D1R neurons exacerbated pain-like (nociceptive) responses. The inconsistent findings may be due to different frequency stimulations. Previous study has shown that distinct frequency stimulations induced the frequency-dependent effects on cell firing in subthalamic nucleus and substantia nigra pars reticulata (Milosevic et al., 2018). Specifically, subthalamic firing attenuated with ≥ 20 Hz stimulation (silenced at 100 Hz), while substantia nigra pars reticulata decreased with ≥ 30 Hz (silenced at 50 Hz) (Milosevic et al., 2018). Consistently, low frequency stimulations (**Figure R2a-c**; 5Hz, 10Hz, and 20Hz) could increase the firing rate of NAc D1R neurons, while high frequency stimulations (**Figure R2d-e**; 30Hz and 100Hz) could inhibit the firing rate of these neurons. Distinct frequency-dependent effects on the firings of NAc D1R neurons may cause different behavioral results between Sato's study and the current study.

Figure R2. Frequency-dependent responses of firing of NAc D1 neurons. (a-e) Raster plot (top) and PSTH (bottom) of the NAc D1 unit as in showing the firing activity before, during, and after 1 s optical stimulation (5Hz, 10Hz, 20Hz, 30Hz, and 100Hz). Blue ticks indicate 473 nm laser pulses.

Comment 3a: When the authors are talking about the NAc D1 neurons, I strongly feel that they are avoiding exposing that NAc D2 neurons are also involved in sleep and pain regulation. The authors reported that only ~20% of D2 neurons are labeled by chronic pain, which is not well consistent with a previous study that chronic pain dramatically increased the AMPA/NMDA ratio in NAc D2 neurons (Ren, 2016), indicating an increase of D2 neuronal activity. Furthermore, 20% of neurons can play an important role in regulating both sleep and pain because activation of D2 neurons increases sleep (Oishi, 2017). Chronic pain activates D2 neurons and D2-neuron activation increases sleep are NOT in the context of chronic pain-induced sleep disorders. The author should carefully address this gap but not avoid talking about these issues.

Response 3: We appreciate the reviewer's insight. Ren and his colleagues found the dramatically increase of the AMPA/NMDA ratio in NAc D2 neurons by ex vivo brain slices recording in SNI mice (Ren et al., 2016). In the same study, they also found the mEPSC frequency of NAc D2 neurons decreased in SNI mice (Ren et al., 2016). Beside, the increase of AMPA/NMDA ratio may not completely contribute to the increased firing and c-fos expressing in NAc D2 neurons in vivo.

To follow reviewer's request, we performed optogenetic activation on D2 neurons specifically in NAc ensemble from c-fos-tTA-Tg mice (**Figure R3**). Similar with the experimental procedure in optrode recordings in **Figure R7** (**Please also see our responses to minor comment 1**), we used the "Tet-Off" and "Cre-On" combination. This involves injecting 2 AAVs (AAV-Tre3G-CRE-WPRE-pA and rAAV-D2-DIO-hChR2-mCherry-WPREs) into NAc of c-Fos-tTA mice two weeks before the CCI surgery, such that ChR2-mCherry was expressed selectively in c-Fos-D2 neurons (NAc-ensemble-D2 neurons) (**Figure R3a-b**). Figure R7 indicates that NAc-ensemble-D2 Neurons are preferentially active during wakefulness. Here, we find that 10-Hz optogenetic stimulation of the NAc-ensemble-D2 neurons increases wakefulness and decreases NREM sleep (**Figure R3c-d**). On the other hand, the same optogenetic stimulation of

these NAc-ensemble-D2 neurons decreases in both the PWL and PWT in CCI mice after 21 days of CCI (**Figure R3e-f**). Oishi et al demonstrated that chemogenetic or optogenetic activation of adenosine A2A receptor-expressing neurons (D2 neurons) in NAc induced NREM sleep (Oishi et al., 2017). Consistently, we optogenetically activated overall NAc-D2 neurons in D2-Cre mice, this activation increased NREM sleep and decreased wakefulness (**Figure R6c**, Please also see our responses to comment 5). Moreover, the same activation rescued both the PWL and PWT in CCI mice (**Figure R6a-b**). These results (activation of overall NAc-D2 neurons) seem to contradict the activation of NAc-ensemble-D2 neurons. However, Chen et al proved that the NAc-D2 MSNs could be classified 27 D2 subtypes by using single-cell RNA sequencing (Chen et al., 2021). Each neuron subtype may have different neural function. These D2 neuron subtypes may contribute to functional heterogeneity on nociceptive responses and sleep-wakefulness. Therefore, it would be interesting to investigate the molecular features of NAc-D2 neuron subtypes enabled us to link different functions to different neuron substrates in future study.

Figure R3. Optogenetic activation of NAc-ensemble-D2 Neurons reduces both NREM sleep and pain thresholds. *a* Schematics showing the labeling and optogenetic activation of NAc-ensemble-D2 Neurons in *c-fos-tTA* transgenic mice underwent CCI surgery. *b* The strategy for specifically labelling D2 neurons in NAc-ensemble neurons by

combinating “Tet-Off” with “Cre-On”. This involves injecting 2 AAVs (AAV-Tre3G-CRE-WPRE-pA and rAAV-D2-DIO-hChR2-mCherry-WPREs) into NAc of *c-Fos-tTA* mice. *c* Representative EEG spectrogram (top), relative EMG trace (middle), and brain states (bottom) from a NAc-ensemble-D2-ChR2 mouse during baseline conditions and blue laser activation of NAc ensemble. Blue stripe indicates laser stimulation (473 nm, 10 Hz, 120 s). *d* Percentage of time in different brain states before, during, and after blue laser activation of NAc-ensemble-D2 Neurons (473 nm, 10 Hz, 120 s). Note the dramatical decrease in NREM sleep ($p < 0.0001$, bootstrap), increase in wakefulness ($p < 0.0001$) no obvious change in REM sleep during laser stimulation ($p > 0.05$), $n = 13$ mice. *e, f* Optogenetic activation (473 nm, 10 Hz) of NAc-ensemble-D2 Neurons significantly decreases thermal PWT ($n = 13$ mice, $F(1.605, 19.26) = 61.19$, $p < 0.0001$, BL vs. Laser, $p < 0.0001$, Pre vs. Laser, $p = 0.001$, Post vs. Laser, $p < 0.0001$) and mechanical PWT ($n = 13$ mice, $F(1.749, 20.99) = 90.74$, $p < 0.0001$, BL vs. Laser, $p < 0.0001$, Pre vs. Laser, $p = 0.0003$, Post vs. Laser, $p = 0.0036$) in *c-fos-tTA-Tg* mice. Repeated measures one-way ANOVA test with Bonferroni's multiple comparisons test for (e, f).

Comment 4a: The authors tried to explain NAc D1R neurons control chronic pain via VTA projection, but emphasize that this pathway is not involved in regulating wakefulness. This is also out of my understanding. Do the pain-labeled D1 neurons significantly different from other NAc D1 neurons because the previous study showed that NAc D1-VTA projection promotes wakefulness (Luo, 2018). If yes, it is necessary to show the difference.

Response 4a: We appreciate the insightful issue. Luo et al did optogenetic activation of NAc D1 neuron terminals in VTA to increase wakefulness (Luo et al., 2018). However, direct inhibiting VTA neurons that received the NAc ensemble inputs did not affect sleep and wakefulness in our study. These different results may be due to manipulating different neuronal populations via different neuronal pathways. Terminal activation has been proved to have adverse side effects (Zingg et al., 2017). Given that a brain region “X” is known to mediate a

behavior/function of interest, determining which neural pathways downstream of X mediate this behavior/function remains challenging (**Figure R4a-b**). Neurons in X project to multiple target nuclei ("Y"). To map the relevant downstream circuit, conventional method relies on activation of ChR2-expressing X axon terminals in a given target nucleus. This may result in unwanted activation of collateral targets via antidromic stimulation (marked by dash lines). Optogenetic activation of NAc D1 neuron terminals in VTA may generate "antidromic spikes" in NAc and other projections. To prove that, we did tetrode recording at the soma of NAc D1 neurons and POA neurons, simultaneously applied laser activations of NAc D1 neurons terminals in VTA (**Figure R4c-d**). We find "antidromic spikes" in the soma of NAc D1 neurons (**Figure R4c**). POA neuron is inhibited during laser stimulations (**Figure R4d**). These "antidromic spikes" propagate to the terminals in POA and inactivate POA neurons, eventually induce wakefulness.

Figure R4. Disadvantages of terminal activation and advantages of transsynaptic activation. *a* A brain region “X” is known to mediate a behavior/function of interest. Neurons in X project to multiple target nuclei (“Y”). To map the relevant downstream circuit, conventional method relies on activation of Chr2-expressing X axon terminals in a given target nucleus. This may result in unwanted activation of collateral targets via antidromic stimulation (marked by dash lines). *b* A virus capable of anterograde transsynaptic spread would allow direct activation of postsynaptic cells in a target region that specifically receive input from region X, by enabling Cre-dependent transgene expression (green) in a Y nucleus (modified from Zingg et al., 2017). *c* Top: Schematic drawing of configuration for optogenetic activation of NAc D1 neuron terminals in VTA and tetra recording at the soma of D1 neurons in NAc. Bottom: Raster plot and PSTH of

recording at the soma of a NAc D1 neuron showing the firing activity before, during, and after 10Hz terminal activation in VTA. d Top: Schematic drawing of configuration for optogenetic activation of NAc D1 neuron terminals in VTA and tetrode recording at the soma of POA neurons. Bottom: Raster plot and PSTH of recording at the soma of a POA neuron showing the firing activity before, during, and after 10Hz terminal activation in VTA.

Moreover, to follow reviewer's request in minor comment 5, we also performed optogenetic activation of NAc ensemble to POA and VTA projections (**Figure R5**). Axon terminals activations in POA and VTA have similar effects on nociceptive responses and sleep-wakefulness states. Optogenetic activation of POA and VTA projections innervated by NAc ensemble reduces both pain thresholds and NREM sleep. Consistently, there is a growing body of evidence suggesting that the soma activations and axon terminals activations have similar effects on sleep-wakefulness states. Optogenetic activation of D1 neuron both at somas in NAc and at axon terminals in VTA or LH induced wakefulness (Luo et al., 2018). Similarly, optogenetic activation of VTA^{Vglut2} neurons both at somas in VTA and at projections in NAc or LH promoted wakefulness (Yu et al., 2019). Consistently, laser activation of CALCA (calcitonin gene-related peptide alpha) or CCK (cholecystokinin) neurons both at cell bodies in the pIII (periculomotor) region and at axon terminals in POA or GiV increased NREM sleep (Zhang et al., 2019).

Therefore, to avoid "antidromic spikes" and specifically inactivate NAc-ensemble-innervating VTA neurons, we applied anterograde trans-synaptic strategy to transsynaptic inhibit VTA neurons that received the NAc ensemble inputs in our study (**Figure R4 and Figure 7**).

Figure R5. Optogenetic activation of NAc ensemble to POA or VTA projections

reduces both pain thresholds and NREM sleep. *a, b* Optogenetic activation (473

nm, 10 Hz) of NAc ensemble to POA projections significantly decreases thermal PWT ($n =$

8 mice, $F(2.202, 15.41) = 54.13$, $p < 0.0001$, BL vs. Laser, $p = 0.0001$, Pre vs. Laser, $p =$

0.041, Post vs. Laser, $p = 0.0152$) and mechanical PWT ($n = 8$ mice, $F(1.604, 11.23) =$

40.59, $p < 0.0001$, BL vs. Laser, $p = 0.0008$, Pre vs. Laser, $p = 0.0213$, Post vs. Laser, $p =$

0.0118) in *c-fos-tTA-Tg* mice. *c* Percentage of time in different brain states before, during,

and after blue laser activation of NAc ensemble to POA projections (473 nm, 10 Hz, 120 s;

$n = 8$ mice). Note the dramatical decrease in NREM sleep ($p < 0.0001$, bootstrap), increase

in wakefulness ($p < 0.0001$) no obvious change in REM sleep during laser stimulation ($p >$

0.05), $n = 8$ mice. *d, e* Optogenetic activation (473 nm, 10 Hz) of NAc ensemble to VTA

projections dramatically decreases thermal PWT ($n = 9$ mice, $F(1.821, 14.57) = 104.8$, $p <$

0.0001, BL vs. Laser, $p < 0.0001$, Pre vs. Laser, $p < 0.001$, Post vs. Laser, $p < 0.001$)

and mechanical PWT ($n = 9$ mice, $F(1.385, 11.08) = 73.15$, $p < 0.0001$, BL vs. Laser, $p <$

0.0001, Pre vs. Laser, $p < 0.001$, Post vs. Laser, $p < 0.001$) in *c-fos-tTA-Tg* mice. *c* Percentage of time in different brain states before, during, and after blue laser activation of NAc ensemble to VTA projections (473 nm, 10 Hz, 120 s; $n = 9$ mice). Note the significant decrease in NREM sleep ($p < 0.0001$, bootstrap), increase in wakefulness ($p < 0.0001$) no obvious change in REM sleep during laser stimulation ($p > 0.05$), $n = 9$ mice. Repeated measures one-way ANOVA test with Bonferroni's multiple comparisons test for (a, b, d, f).

Comment 4b: *In addition, whether optogenetic activation of NAc-VTA projection increases the activity of VTA dopaminergic and glutamatergic neurons? It's better to check the activity of these neurons after optogenetic activation to further explain the downstream circuit mechanisms.*

Response 4b: We thank reviewer for the insightful question. A lot of studies have investigated NAc-VTA projection. NAc D1 neurons mainly targeted VTA non-DA neurons (GABAergic and Glutamatergic neurons) and sparsely targeted VTA DA neurons (Edwards et al., 2017; Luo et al., 2018; Xia et al., 2011). Decreased firing rates and evoked inhibitory postsynaptic currents were recorded in VTA non-DA neurons when photostimulations activated axonal terminals of NAc D1R neurons (Luo et al., 2018). Moreover, VTA^{Vgat} neurons promoted NREM sleep, while VTA^{Vglut2} neurons promoted wakefulness (Yu et al., 2019). Inhibiting VTA^{Vgat} and VTA^{Vglut2} neurons that received the NAc ensemble inputs has opposite effects on sleep and wakefulness, and may not affect sleep and wakefulness. In our study, to avoid “antidromic spikes”, we did not apply optogenetic activation of NAc-VTA projection, but specifically inactivate NAc-ensemble-innervating VTA neurons. In future study, it would be interesting to investigate whether and how “optogenetic activation of NAc-VTA projection increases the activity of VTA dopaminergic neurons.”

Comment 5: Same experiments in Fig.5 are suggested to perform in NAc D2 neurons.

Response 5: We thank the reviewer for the good suggestion. We optogenetically (473 nm, 10 Hz) activated D2 neurons in D2-Cre mice with Cre-dependent expression of ChR2 in the NAc. This activation rescued both the PWL and PWT in CCI mice (**Figure R6a-b**). Moreover, optogenetic stimulation (10 Hz x 2 min) of NAc D2 neurons increased the percentage of NREM sleep, and decreased wakefulness and REM sleep, resulting in an overall upregulation of NREM sleep (**Figure R6c**).

Figure R6. Optogenetic activation of NAc D2 neurons rescues pain thresholds and promotes NREM sleep. *a, b* Optogenetic activation (473 nm, 10 Hz) of NAc D2 neurons significantly increases thermal PWL ($n = 13$ mice, $F(2.46, 29.5) = 54.3$, $p < 0.0001$, BL vs. Laser, $p = 0.73$, Pre vs. Laser, $p < 0.001$, Post vs. Laser, $p < 0.001$) and mechanical PWT ($n = 13$ mice, $F(2.06, 24.8) = 49.6$, $p < 0.0001$, BL vs. Laser, $p = 0.735$, Pre vs. Laser, $p < 0.001$, Post vs. Laser, $p < 0.001$) to the baseline levels in D2-cre mice. *c* Percentage of time in different brain states before, during, and after blue laser activation of NAc D2 neurons (473 nm, 10 Hz, 120 s; $n = 13$ mice). Note the significant increase in NREM sleep ($p < 0.0001$, bootstrap) and decrease in wakefulness ($p < 0.0001$) and REM ($p < 0.001$) sleep during laser stimulation. Repeated measures one-way ANOVA test with Bonferroni's multiple comparisons test for (*a, b*).

Minor

Minor comment 1: *Are there any D2 neurons labeled in Fig1B also activated wakefulness? Maybe the author can present D1 and D2 neurons activity across sleep-wake transitions in a supplementary figure.*

Response 1: We appreciate the helpful suggestion from reviewer. To follow reviewer's request, similar to the experiment in the responses to major comment 3b, we performed optrode recording on D2 neurons specifically in NAc ensemble from c-fos-tTA-Tg mice (Figure R7a-b). We combined "Tet-Off" with "Cre-On" by injecting 2 AAVs (AAV-Tre3G-CRE-WPRE-pA and rAAV-D2-DIO-hChR2-mCherry-WPREs) into c-Fos-tTA mice two weeks before the CCI surgery. Consequently, ChR2-mCherry was expressed selectively in c-Fos-D2 neurons (NAc-ensemble-D2) (Figure R7a-b). Using the same criteria as in Fig 1, we screened and identified the in vivo ChR2-expressing NAc-ensemble-D2 neurons through single-unit recording (**Figure R7c-d**). Next, we monitored the spike firing of NAc-ensemble-D2 neurons across sleep and wake states. In CCI mice, the mean spiking rate of NAc-ensemble-D2 neurons was considerably higher during wakefulness compared to either NREM or REM sleep (Figure R7e-f). Furthermore, the spiking rate of these NAc-ensemble-D2 neurons was similar with NAc-ensemble neurons in CCI mice during wakefulness or NREM/REM sleep (Figure R7f). We also quantified the brain-state preference of recorded neurons by calculating their wake-NREM modulation $[(R_{\text{wake}} - R_{\text{NREM}})/(R_{\text{wake}} + R_{\text{NREM}})]$, R is the averaged firing rate within each brain state] and REM-NREM modulation $[(R_{\text{REM}} - R_{\text{NREM}})/(R_{\text{REM}} + R_{\text{NREM}})]$ (**Figure R7g**). NAc-ensemble-D2 and NAc-ensemble neurons showed similar brain-state preference. Furthermore, we analyzed the changes in the firing rate of NAc-ensemble-D2 neurons during the transitions between two different brain states. The averaged firing rates of NAc-ensemble-D2 neurons greatly increased before the onset of wakefulness during the transitions from NREM/REM sleep to wakefulness and gradually decreased during transitions from wakefulness to NREM sleep (**Figure R7h**). No obvious change was observed during the

transition from NREM to REM sleep. NAc-ensemble-D2 and NAc-ensemble neurons exhibited similar changes in the firing rate during the transitions between two different brain states. All these results suggest NAc-ensemble-D2 Neurons are preferentially active during wakefulness. Moreover, Luo et al demonstrated that NAc-D1 neurons activity increased before NREM to wake transitions and decreased before wake to NREM transitions (Luo et al., 2018). In our study, about 80% NAc-ensemble are D1 neurons, the changes of their activity across sleep-wake transitions (**Figure 1m**) are consistent with previous study.

Figure R7. Nac-ensemble-D2 Neurons are preferentially active during wakefulness. a

Schematics showing the labeling and recording of NAc-ensemble-D2 Neurons in c-fos-tTA transgenic mice underwent CCI surgery. b The strategy for specifically labelling D2 neurons in NAc-ensemble neurons by combining “Tet-Off” with “Cre-On”. This involves injecting 2 AAVs (AAV-Tre3G-CRE-WPRE-pA and rAAV-D2-DIO-hChR2-mCherry-WPREs) into NAc of c-Fos-tTA mice. c Waveforms of average spontaneous (red) and

individual laser-evoked (blue) spikes from an identified NAc-ensemble-D2 neuron in the NAc ensemble. **d** Example of raster plot from a NAc-ensemble-D2 neuron showing consecutive laser stimulation trials at 10 Hz. **e** Representative firing rate of a NAc-ensemble-D2 neuron (bottom) together with brain states (color-coded), EMG trace (middle), and EEG spectrogram (top). **f** Violin plot displaying the individual firing rates of identified NAc-ensemble neurons (blue, $n = 28$ units from 6 mice, the same data as in Fig. 1k) and NAc-ensemble-D2 neurons in CCI mice (red, $n = 29$ units from 5 mice) during different brain states. *** $p < 0.001$ (State, $F_{(2,110)} = 158.7$, $p < 0.001$, two-way ANOVA; wake vs. NREM, $p < 0.001$; wake vs. REM, $p < 0.001$, Bonferroni's multiple comparisons test). ns: not significant, versus the NAc-ensemble neurons during different brain states (Group, $F_{(1,55)} = 0.557$, $p = 0.459$, two-way ANOVA; NAc-ensemble vs. NAc-ensemble-D2, wake: $p > 0.99$, NREM: $p = 0.351$, REM: $p > 0.99$; Bonferroni's multiple comparisons test). Data are presented as median (red or yellow line) with 25th and 75th percentile (dash line). **g** The distributions of both Wake-NREM and REM-NREM modulations were similar between NAc-ensemble neurons and NAc-ensemble-D2 neurons. **h** Mean firing rates of identified NAc-ensemble-D2 neurons (blue, $n = 29$ units from 5 mice) during different brain state transitions.

Minor comment 2: Figure 1F & 2D, a significant 10 Hz signal in EEG heatmap. Is it from opto-stimulation? However Figure 1L did not show it. Noise elimination is needed to increase the reliability of results.

Response 2: We thank the reviewer for pointing out this issue. We think the 10 Hz signal in EEG heatmap in Figure 2f (not Figure 1F & 2D) may not be noise and does not result from opto-stimulation. If the “10Hz-noise” result from opto-stimulation, it may also show up in Figure 2l and Supplementary Figure 5d, since we used same laser stimulation as in Figure 2f. The finding from current study is consistent with several prior studies. Photostimulation in NAc-D1 neurons at 20Hz could induce a peak at 20Hz in EEG spectral power in D1-ChR2-mCherry mice, but not in D1-mCherry mice (Figure R8a)(Luo et al., 2018). Optogenetic inhibiting the occipital cortex at 20 Hz promoted a significant 20 Hz signal in

EEG spectrogram (Figure R8b)(Wang et al., 2022). Optogenetic activating the prefrontal parvalbumin (PV) interneurons and somatostatin (SST) interneurons at 40 Hz induced a clear peak at 40Hz in EEG spectral power of PV-cre and SST-cre mice, respectively (Figure R8c)(Liu et al., 2020b). Laser stimulation at 40Hz in PV neuron in the basal forebrain evoked a significant 40 Hz signal in EEG spectrogram. The power of 40 Hz signal in EEG spectrogram decreased obviously after vitamin D-deficient diet (Figure R8d) (Yu et al., 2022). In these studies, the targeted neurons of different brain regions might have a close relationship with cerebral cortex (EEG recording location). Photo-stimulations in these targeted neurons could induced signal at stimulated frequency in cerebral cortex.

Figure R8. Optogenetic stimulations in different brain regions induce signal at stimulated frequency in EEG. *a* EEG power before (red trace) and after (blue trace) onset of photo-stimulation in NAc of D1-ChR2-mCherry (Left) or D1-mCherry (Right) mice. 20 Hz optogenetic activation of NAc D1 neurons induced a peak at 20Hz in EEG spectral power (modified from Luo et al., 2018). *b* Schematic of experiment showing 20 Hz optogenetic inhibition of the occipital cortex and a significant 20 Hz signal in EEG

spectrogram (indicated by red arrows)(modified from Wang et al., 2022). *c* Example spectral power of EEG with (blue) and without (black) light stimuli of prefrontal parvalbumin interneurons (Left) and somatostatin interneurons (Right) at 40 Hz. A clear peak (indicated by red arrow) at 40Hz observed in EEG spectral power (modified from Liu et al., 2020). *d* Representative time-frequency spectrogram of EEG from before and after vitamin D-deficient diet (0 wk, left; 6 wk, right). EEG spectrogram before, during, and after 40 Hz (0.5 s) laser stimulation in the basal forebrain. Note a significant 40 Hz signal in EEG spectrogram (indicated by red arrows)(modified from Yu et al., 2022).

Minor comment 3: *In Figure S3, the firing rate of NAc D1 neurons was not altered in sleep-wake transitions in naïve mice, do the author think these neurons are not involved in the regulation of wakefulness? How to understand the increased calcium activity of NAc neurons in wakefulness from the previous study?*

Response 3: We appreciate the reviewer's insight. In Figure S3, we are not sure whether the labelled neurons in sham mice were D1 neurons. In sham mice, we used the same tTA-TRE (Tet-Off)-based viral system in c-Fos-tTA mice as CCI mice (Figure 1a). During 3 weeks after sham surgery, NAc neurons may be activated and tagged with ChR2 by feeding, social, locomotion, or sleep-wake behaviors (Luo et al., 2018; O'Connor et al., 2015; Oishi et al., 2017; Williams et al., 2020; Zhu et al., 2016). Consistently, some identified neurons in sham mice were not typical sleep- or wake-active neurons (Figure 1k-l). Moreover, the mean firing rate of these identified neurons in sham mice was comparable during wakefulness, NREM, and REM sleep (Figure 1k). These identified neurons may be related with feeding, social, or locomotion behaviors. The mean firing rate of these neurons may be not altered in sleep-wake transitions. **Please also see our responses to major comment 1.**

Minor comment 4: *Figure 6H, why these VTA and POA-projecting cells are normalized to total VTA and POA cells, but not NAc cells?*

Response 4: We apologize if we did not make clear statement in the Figure 6 legend. In

Figure 6h of the initial submission, VTA and POA-projecting NAc ensemble neurons are normalized to all VTA and POA-projecting neurons, but not all VTA and POA cell. These calculations may help us better evaluate the important roles of NAc-ensemble to POA and to VTA pathways in regulating pain and sleep. Moreover, we really appreciate the helpful suggestion from reviewer. To follow the reviewer's request, we have reanalyzed data. VTA and POA-projecting NAc ensemble neurons are normalized to all NAc ensemble neurons (POA+NAc-ensemble+ / NAc-ensemble+ and VTA+NAc-ensemble+ / NAc-ensemble+) in new Figure 6h. **Please also see our responses to comment 6 from reviewer #1.**

Minor comment 5: Inhibition experiments in Figure 7 are indirect evidence. Whether the stimulation of NAc to POA and VTA projections will get similar results?

Response 5: To follow reviewer's request, we performed optogenetic activation of NAc ensemble to POA and VTA projections (**Figure R5**). **Please see also our responses to major comment 4.** Optogenetic activation of POA and VTA projections innervated by NAc ensemble reduces both pain thresholds and NREM sleep. These similar results may be due to manipulating same neuronal populations via same neuronal pathways. Optogenetic activation of NAc ensemble neuron terminals in VTA may generate "antidromic spikes" in NAc and other projections. These "antidromic spikes" propagate to the terminals in POA and inactivate POA neurons, eventually reduce NREM sleep. Similarly, optogenetic activation of the NAc-ensemble terminals in POA may also generate "antidromic spikes" in NAc and VTA projections and inactivate VTA neurons, eventually reduce pain thresholds. All these suggest optogenetic activating of NAc-ensemble-innervating POA and VTA projections respectively have similar effect on nociceptive responses and sleep-wakefulness states. We also display the Figure R5 data as following:

Figure R5. Optogenetic activation of NAc ensemble to POA or VTA projections

reduces both pain thresholds and NREM sleep. *a, b* Optogenetic activation (473 nm, 10 Hz) of NAc ensemble to POA projections significantly decreases thermal PWL ($n = 8$ mice, $F(2.202, 15.41) = 54.13$, $p < 0.0001$, BL vs. Laser, $p = 0.0001$, Pre vs. Laser, $p = 0.041$, Post vs. Laser, $p = 0.0152$) and mechanical PWT ($n = 8$ mice, $F(1.604, 11.23) = 40.59$, $p < 0.0001$, BL vs. Laser, $p = 0.0008$, Pre vs. Laser, $p = 0.0213$, Post vs. Laser, $p = 0.0118$) in *c-fos-tTA-Tg* mice. *c* Percentage of time in different brain states before, during, and after blue laser activation of NAc ensemble to POA projections (473 nm, 10 Hz, 120 s; $n = 8$ mice). Note the dramatical decrease in NREM sleep ($p < 0.0001$, bootstrap), increase in wakefulness ($p < 0.0001$) no obvious change in REM sleep during laser stimulation ($p > 0.05$), $n = 8$ mice. *d, e* Optogenetic activation (473 nm, 10 Hz) of NAc ensemble to VTA projections dramatically decreases thermal PWL ($n = 9$ mice, $F(1.821, 14.57) = 104.8$, $p < 0.0001$, BL vs. Laser, $p < 0.0001$, Pre vs. Laser, $p < 0.001$, Post vs. Laser, $p < 0.001$) and mechanical PWT ($n = 9$ mice, $F(1.385, 11.08) = 73.15$, $p < 0.0001$, BL vs. Laser, $p <$

0.0001, Pre vs. Laser, $p < 0.001$, Post vs. Laser, $p < 0.001$) in *c-fos-tTA-Tg* mice. **c**
Percentage of time in different brain states before, during, and after blue laser activation
of NAc ensemble to VTA projections (473 nm, 10 Hz, 120 s; $n = 9$ mice). Note the significant
decrease in NREM sleep ($p < 0.0001$, bootstrap), increase in wakefulness ($p < 0.0001$) no
obvious change in REM sleep during laser stimulation ($p > 0.05$), $n = 9$ mice. Repeated
measures one-way ANOVA test with Bonferroni's multiple comparisons test for (a, b, d, f).

Reviewer #3 (Remarks to the Author):

This is an elegant study that significantly advances our knowledge about the interplay between sleep and pain. I will summarize the main data to let the authors know what I believe they should consider before publication. Sun and cols. used c-fos-tTA transgenic mice to express several AAV-carried effect genes under the c-fos promoter, so that the transgenes were preferentially expressed in neurons with high expression of c-Fos in the target region. The authors focused on the Nucleus accumbens (NAc) an interface region in the control of movement, pain, and sleep-wake cycle. The expression of the controlled transgenes was released (by terminating doxycycline supply) between day 0 and 21 after CCI (chronic constriction of the sciatic nerve) or CFA (Complete Freund's adjuvant) injections. So, NAc neurons activated in these chronic pain models were targeted. These neurons showed naturally increased activity during nociceptive tests and wakefulness (with decreased activity during non-REM sleep (NREM sleep)) (Fig 1). The optogenetic stimulation of these neurons further decreases mechanical paw withdrawal threshold (PWT) and thermal latency (PWL); and increases wakefulness (decreasing NREM sleep) in CCI or CFA animals (Fig 2 and 3). Complementary, their optogenetic inactivation induces the opposite effects (Fig 4). The majority (80%) of CCI target neurons express dopamine D1 receptors and optogenetic stimulation of these D1 neurons (using D1-Cre mice) decreased both the PWT and PWL and increased the percentage of wakefulness (decreasing sleep); while their optogenetic inactivation increases the PWT and PWL and increases sleep (Fig 5). Anterograde tracing of NAc target neurons identified several downstream brain regions that are known to regulate sleep/wakefulness and/or nociception and a double-labeling assay suggest that these neurons are GABAergic principal medium spiny neurons. The authors focused in VTA (ventral tegmental area) and POA (preoptic area) and used a retrogradely label strategy to demonstrate that a portion (30%) of NAc target neurons divergently projected to both POA and VTA (fig 06). Finally, the authors used a dual-viral system to optogenetically inactivate the VTA or POA neurons innervated by the NAc targeted neurons. Inactivation of POA neurons decreases sleep (with no effect on nociceptive tests), while inactivation of VTA neurons decreases PWT and PWL (with no effect on sleep) (fig 07). Based on this broad set of data, the authors concluded that these NAc-targeted neurons form "... a common NAc ensemble that encodes chronic pain and controls sleep".

The bidirectional relationship between negative changes in sleep and pain has been the focus of several studies in the last two decades and the NAc is emerging as a key candidate to mediate the effect of decreased sleep on pain processing. However, the underlying mechanisms are poorly understood, and this is the first study to identify a neural population in the NAc that is apparently involved in the control of pain and sleep/wake cycle. These data will impact the literature on the field and are at the edge of knowledge. I have one main concern about the experiments and some suggestions to improve the manuscript reading and data presentation.

Response : We appreciate the reviewer for your positive comments on our work.

Major

Comment 1a: *How are the authors sure that the NAc targeted neurons does not simply control the movement of paw withdrawal after mechanical or thermal stimulation in CCI or CFA animals? This is important not only in view of the role of NAc (especially D1 neurons) in initiating movement, but also because paw withdrawal was consistently evoked throughout the experiments with no apparent need, since these behavioral data (days 3 to 21) were not shown. Indeed, it is unfortunate that the authors did not perform a single experiment to assess motor function (another than PW) upon targeted neurons manipulation. I believe that the absence of PW changes in sham animals does not solve the problem because they are not under neuropathic (CCI) or inflammatory (CFA) sensitization. Some kind of motor function monitoring would also be welcome during optogenetic stimulation that induces the transition from sleep to wakefulness. This reviewer would like to see at least one nociceptive experiment performed without motor bias. Conditioned place preference (CPP) under target neurons inhibition would be a great alternative and it is feasible in CCI model.*

Response 1a: We appreciate the helpful suggestion from the reviewer. To follow the reviewer's request, we did EMG electrodes implantation in MG muscle (Medial gastrocnemius) of hindlimb in NAc-ensemble-ChR2 and NAc-ensemble-eNpHR mice (**Figure. R9a**). The EMG signals could be detected when the movement of paw withdrawal was induced by mechanical or thermal

stimulation in CCI mice (**Figure. R9b**). Next, both optogenetic activation and inactivation were applied to NAc ensemble neurons in NAc-ensemble-ChR2 and NAc-ensemble-eNpHR mice respectively, we did not find the significant change in EMG spectral power during blue or yellow laser stimulations (**Figure. R9c-9f**) under free moving state. The reviewer kindly suggested to perform CPP task under target neurons inhibition. Since the mice had an optical fiber cable connected with laser, they could not move freely through the sliding door between two chambers in CPP task. We performed open field test so that the mice with optical fiber cables could move freely in a large chamber. The total distance traveled by NAc-ensemble-ChR2 (or NAc-ensemble-eNpHR) mice was compared during laser off and on period. Both blue and yellow laser stimulations have no effect on the total distance traveled by NAc-ensemble-ChR2 and NAc-ensemble-eNpHR mice, respectively (**Figure R9g-9h**).

Figure R9. EMG recording of hindlimb muscles and optogenetic stimulation of NAc

ensemble. a Illustration of intramuscular electrode implantation on MG and the head connector. Dash lines represented the EMG cables. **b** Representative traces showing EMG recording of hindlimb muscles in a CCI mouse before, during, and after a 0.07-g von Frey filament stimuli (top) and thermal stimuli (bottom). **c** EMG trace recording of hindlimb muscles from a CCI mouse during baseline conditions and 10 Hz blue laser (473 nm, 120 s) activation of NAc ensemble neurons. **d** Normalized EMG spectrogram aligned to blue laser stimulation time (During 3-hour recording, blue laser stimulations were delivered randomly from a uniform distribution between 4 and 10 min). Note no significant change in EMG spectral power upon blue laser stimulation. **e** EMG trace recording of hindlimb muscles from a CCI mouse during baseline conditions and yellow laser (589 nm, 8 s on/2 s off, 120 s) inactivation of NAc ensemble neurons. **f** Normalized EMG spectrogram aligned to yellow laser stimulation time (During 3-hour recording, yellow laser stimulations were delivered randomly from a uniform distribution between 4 and 10 min). Note no obvious change in EMG spectral power upon yellow laser stimulation. **g** Left: Representative heat maps of NAc-ensemble-ChR2 mice within the open field box over a 10-minute period in the absence or presence of blue laser activation. Right: Optogenetic activation of NAc ensemble did not change the total distance traveled by NAc-ensemble-ChR2 mice in the open field test. **h** Left: Representative heat maps of NAc-ensemble-eNpHR mice within the open field box over a 10-minute period in the absence or presence of yellow laser inactivation. Right: Optogenetic inactivation of NAc ensemble did not change the total distance traveled by NAc-ensemble- eNpHR mice in the open field test.

Comment 1b: The experimental strategy is based on targeting NAc neurons that express *c-Fos* after CCI or CFA, but there is no comparison of the expression of *c-Fos* (or associated fluorescent proteins) between CCI/CFA and sham animals. Such a comparison would be welcome to assess the effect of CCI/CFA on NAc activity.

Response 1b: We appreciate the helpful suggestion from reviewer. To follow the reviewer's request, we applied FISH (Fluorescent in situ Hybridization) to compare the expression of *c-Fos* between CCI and sham mice. The *c-Fos* expression level

in NAc three weeks after CCI surgery was dramatically higher than that after sham surgery (**Figure R10**).

Figure R10. The c-Fos expression in NAc three weeks after sham or CCI surgery.

a FISH showing the c-Fos expression in NAc three weeks after sham and CCI surgery respectively. **b** Statistical analysis of c-Fos expression in NAc under different conditions as in (a). ** $p = 0.0022$ versus the c-Fos+ in sham mice (Mann Whitney test).

Minor

Minor comment 1: Abstract: The chronic pain models used should be mentioned in the abstract.

Response 1: According to reviewer's suggestion, we add the chronic pain models in the abstract on Page 2, line 4-5.

Results:

Minor comment 2: Please use the same scale on the Y-axis for similar experiments throughout the presentation of results. The lack of this standardization makes it difficult to compare data between different figures.

Response 2: We thank the reviewer for the suggestion. We set the same scale on the Y axis for similar experiments. For example, the same maximum and interval on the Y axis are used in thermal PWL and mechanical PWT tests in new Figs. 2c-d, 2i-j, 3b-c, 4b-c, 5d-e, 7d-e, 7j-k, and Supplementary Figs. S1b-c, S5a-b, S7b-

c, S8b-c, S9a-b, S10a-b, S10d-e.

Minor comment 3: *Baseline values for PWT/L should be plotted in figures, as it was done in figure 4 B and C.*

Response 3: We thank the reviewer for the good suggestion. To follow the reviewer's suggestion, we add baseline values for PWT/L in new Figs. 2c-d, 2i-j, and Supplementary Figs. S5a-b, S7b-c, S8b-c, S9a-b, S10a-b, S10d-e and new statistic values in legends correspondingly.

Minor comment 4: *Fig S1 E1-G1 there is no red plot or line.*

Response 4: We apologize for the error and have corrected it.

Please see the detail in Supplementary Materials on Page 2.

Minor comment 5: *Please indicate that FISH means (Fluorescent in situ Hybridization) in the first figure legend*

Response 5: We add Fluorescent in situ Hybridization in the first figure legend.

Methods:

Minor comment 6: *Why were the NAc injections unilateral? Is there any laterality between the injected side (NAc) and the side subjected to CCI/CFA (paw)?*

Response 6: We thank the reviewer for pointing out this issue. Because we performed CCI/CFA unilaterally. AAV virus was injected into the contralateral side to CCI/CFA (paw) that were painful information inputs.

Please see the detail on Page 25, Line 3-4 in the Methods section of the revised version.

Minor comment 7: *There is no mention to mice used in experiments of figure 5 (Cre) and 6 (black)*

Response 7: We clarify the mice used in new Figs. 5a-b, 5d, 5g, 6b,6d,6e, and legends correspondingly.

Discussion

Minor comment 8: “*Our results show that neurons in the dual-functioning ensemble in CCI mice exhibited several cellular dynamics distinct from non-ensemble neurons (Figures 2B and 2H)*” *Figures 2 B and H are not related to these findings.*

Response 8: We regret the error and have corrected it as following: “ Our results show that neurons in the dual-functioning ensemble in CCI mice exhibited several cellular dynamics distinct from non-ensemble neurons (Figs. 1f-i, 1k-m, and Supplementary Fig. 3) ”.

Please see the detail in the Discussion section on Page 18, line 9-10.

References

Chen, R., Blosser, T.R., Djekidel, M.N., Hao, J., Bhattacharjee, A., Chen, W., Tuesta, L.M., Zhuang, X., Zhang, Y., 2021. Decoding molecular and cellular heterogeneity of mouse nucleus accumbens. *Nature Neuroscience* 24, 1757-1771.

Edwards, N.J., Tejada, H.A., Pignatelli, M., Zhang, S., McDevitt, R.A., Wu, J., Bass, C.E., Bettler, B., Morales, M., Bonci, A., 2017. Circuit specificity in the inhibitory architecture of the VTA regulates cocaine-induced behavior. *Nature Neuroscience* 20, 438-448.

Francis, T.C., Chandra, R., Friend, D.M., Finkel, E., Dayrit, G., Miranda, J., Brooks, J.M., Iñiguez, S.D., O'Donnell, P., Kravitz, A., Lobo, M.K., 2015. Nucleus accumbens medium spiny neuron subtypes mediate depression-related outcomes to social defeat stress. *Biological psychiatry* 77, 212-222.

Hunt, S.P., Pini, A., Evan, G., 1987. Induction of c-fos-like protein in spinal cord neurons following sensory stimulation. *Nature* 328, 632-634.

Jiang, S., Wang, Y.S., Zheng, X.X., Zhao, S.L., Wang, Y., Sun, L., Chen, P.H., Zhou, Y., Tin, C., Li, H.L., Sui, J.F., Wu, G.Y., 2022. Itch-specific neurons in the ventrolateral orbital cortex selectively modulate the itch processing. *Sci Adv* 8, eabn4408.

Kai, N., Nishizawa, K., Tsutsui, Y., Ueda, S., Kobayashi, K., 2015. Differential roles of dopamine D1 and D2 receptor-containing neurons of the nucleus accumbens shell in behavioral sensitization. *Journal of neurochemistry* 135, 1232-1241.

Lazarus, M., Shen, H.Y., Cherasse, Y., Qu, W.M., Huang, Z.L., Bass, C.E., Winsky-Sommerer, R., Semba, K., Fredholm, B.B., Boison, D., Hayaishi, O., Urade, Y., Chen, J.F., 2011. Arousal effect of caffeine depends on adenosine A2A receptors in the shell of the nucleus accumbens. *The Journal of neuroscience : the official journal of the Society for Neuroscience* 31, 10067-10075.

Liu, D., Li, W., Ma, C., Zheng, W., Yao, Y., Tso, C.F., Zhong, P., Chen, X., Song, J.H., Choi, W., Paik, S.B., Han, H., Dan, Y., 2020a. A common hub for sleep and motor control in the substantia nigra. *Science (New York, N.Y.)* 367, 440-445.

Liu, L., Xu, H., Wang, J., Li, J., Tian, Y., Zheng, J., He, M., Xu, T.-L., Wu, Z.-Y., Li, X.-M., Duan, S.-M., Xu, H., 2020b. Cell type-specific differential modulation of prefrontal cortical GABAergic interneurons on low gamma rhythm and social interaction. *Science Advances* 6, eaay4073.

Luo, Y.J., Li, Y.D., Wang, L., Yang, S.R., Yuan, X.S., Wang, J., Cherasse, Y., Lazarus, M., Chen, J.F., Qu, W.M., Huang, Z.L., 2018. Nucleus accumbens controls wakefulness by a subpopulation of neurons expressing dopamine D(1) receptors. *Nature communications* 9, 1576.

Milosevic, L., Kalia, S.K., Hodaie, M., Lozano, A.M., Fasano, A., Popovic, M.R., Hutchison, W.D., 2018. Neuronal inhibition and synaptic plasticity of basal ganglia neurons in Parkinson's disease. *Brain : a journal of neurology* 141, 177-190.

O'Connor, E.C., Kremer, Y., Lefort, S., Harada, M., Pascoli, V., Rohner, C., Lüscher, C., 2015. Accumbal D1R Neurons Projecting to Lateral Hypothalamus Authorize Feeding. *Neuron* 88, 553-564.

Oishi, Y., Xu, Q., Wang, L., Zhang, B.J., Takahashi, K., Takata, Y., Luo, Y.J., Cherasse, Y., Schiffmann, S.N., de Kerchove d'Exaerde, A., Urade, Y., Qu, W.M., Huang, Z.L., Lazarus, M., 2017. Slow-wave sleep is controlled by a subset of nucleus accumbens core neurons in mice. *Nature communications* 8, 734.

Ren, W., Centeno, M.V., Berger, S., Wu, Y., Na, X., Liu, X., Kondapalli, J., Apkarian, A.V., Martina, M.,

Surmeier, D.J., 2016. The indirect pathway of the nucleus accumbens shell amplifies neuropathic pain. *Nat Neurosci* 19, 220-222.

Smith, R.J., Lobo, M.K., Spencer, S., Kalivas, P.W., 2013. Cocaine-induced adaptations in D1 and D2 accumbens projection neurons (a dichotomy not necessarily synonymous with direct and indirect pathways). *Curr Opin Neurobiol* 23, 546-552.

Tellez, L.A., Perez, I.O., Simon, S.A., Gutierrez, R., 2012. Transitions between sleep and feeding states in rat ventral striatum neurons. *Journal of neurophysiology* 108, 1739-1751.

Voscopoulos, C., Lema, M., 2010. When does acute pain become chronic? *British Journal of Anaesthesia* 105, i69-i85.

Wang, Z., Fei, X., Liu, X., Wang, Y., Hu, Y., Peng, W., Wang, Y.-w., Zhang, S., Xu, M., 2022. REM sleep is associated with distinct global cortical dynamics and controlled by occipital cortex. *Nature communications* 13, 6896.

Williams, A.V., Duque-Wilckens, N., Ramos-Maciel, S., Campi, K.L., Bhela, S.K., Xu, C.K., Jackson, K., Chini, B., Pesavento, P.A., Trainor, B.C., 2020. Social approach and social vigilance are differentially regulated by oxytocin receptors in the nucleus accumbens. *Neuropsychopharmacology : official publication of the American College of Neuropsychopharmacology* 45, 1423-1430.

Xia, Y., Driscoll, J.R., Wilbrecht, L., Margolis, E.B., Fields, H.L., Hjelmstad, G.O., 2011. Nucleus Accumbens Medium Spiny Neurons Target Non-Dopaminergic Neurons in the Ventral Tegmental Area. *The Journal of Neuroscience* 31, 7811-7816.

Yu, S., Park, M., Kang, J., Lee, E., Jung, J., Kim, T., 2022. Aberrant Gamma-Band Oscillations in Mice with Vitamin D Deficiency: Implications on Schizophrenia and its Cognitive Symptoms. *Journal of personalized medicine* 12.

Yu, X., Li, W., Ma, Y., Tossell, K., Harris, J.J., Harding, E.C., Ba, W., Miracca, G., Wang, D., Li, L., Guo, J., Chen, M., Li, Y., Yustos, R., Vyssotski, A.L., Burdakov, D., Yang, Q., Dong, H., Franks, N.P., Wisden, W., 2019. GABA and glutamate neurons in the VTA regulate sleep and wakefulness. *Nat Neurosci* 22,

106-119.

Zhang, Z., Zhong, P., Hu, F., Barger, Z., Ren, Y., Ding, X., Li, S., Weber, F., Chung, S., Palmiter, R.D., Dan, Y., 2019. An Excitatory Circuit in the Periocolomotor Midbrain for Non-REM Sleep Control. *Cell* 177, 1293-1307.e1216.

Zhou, Y., Zhu, H., Liu, Z., Chen, X., Su, X., Ma, C., Tian, Z., Huang, B., Yan, E., Liu, X., Ma, L., 2019. A ventral CA1 to nucleus accumbens core engram circuit mediates conditioned place preference for cocaine. *Nature Neuroscience* 22, 1986-1999.

Zhu, X., Ottenheimer, D., DiLeone, R.J., 2016. Activity of D1/2 Receptor Expressing Neurons in the Nucleus Accumbens Regulates Running, Locomotion, and Food Intake. *Frontiers in Behavioral Neuroscience* 10.

Zingg, B., Chou, X.-l., Zhang, Z.-g., Mesik, L., Liang, F., Tao, H.W., Zhang, L.I., 2017. AAV-Mediated Anterograde Transsynaptic Tagging: Mapping Corticocollicular Input-Defined Neural Pathways for Defense Behaviors. *Neuron* 93, 33-47.

REVIEWER COMMENTS

Reviewer #1 (Remarks to the Author):

The authors are highly responsive to reviewers' critiques. They have performed extensive additional experiments to validate their conclusions as well as to tease apart the important differences between their findings and previously published results from various other groups. These additional results will be highly informative to the neuroscience community for guiding the proper usage of optogenetics, as well as to better understand the NAc functional neuronal ensembles. They should be all incorporated into the supplementary (or even main) figures rather than only used to show the reviewers. Related results and discussions should be also incorporated into the main text. Detailed suggestions are listed below.

Response #1: This is an important clarification that needs to be incorporated into the main text. A single citation without explanation is not sufficient, as readers outside of the pain field would not immediately understand the c-Fos expression scenario here.

Response #2: "arousal" may be more suitable here than "alertness".

Response #6: Fig. 6H: mathematically, if POA+/total ensemble is ~ 80%, and VTA+/total ensemble is ~ 80%, then the overlapping subpopulation, e.g. POA+VTA+/total ensemble, should be at least 60% ($80\%+80\%-100\%=60\%$). The fact that it was much lower (~30%) is puzzling and needs to be addressed.

Figure R2 demonstrates important in vivo evidence for frequency-dependent activation of NAc neurons by optogenetic stimulations. This can be incorporated into supplementary materials. It will also be helpful to specify the duration of individual laser pulses used here.

Figures R3 and R6 demonstrate the important differences between D2 ensemble neurons and D2 total populations in regulating pain and wakefulness. These should be included in the main and supplementary figures.

Figures R4 and R5 demonstrate the important differences between terminal versus transsynaptic cell body stimulations. This should be included in the supplementary figures.

Figure R7 complements main Figure 1 and should be included in the supplementary figures.

Reviewer #2 (Remarks to the Author):

In the revised manuscript, the authors have well addressed my comments with new experiments, discussion, and illustration of previous data. I do believe that this vision has substantial improvements and seems close to be published. A few minor comments to the authors for their consideration are listed

below.

1. In Comments 1, the authors discussed the heterogeneity of NAc non-ensemble neurons to explain why activation of these non-ensemble did not alter sleep-wake behaviors. However, it is better to tune down this conclusion because activation of PVT- or VTA-projections in the NAc are also wake-promoting (these are all non-specific activation).
2. Comments 2, if the different frequencies of stimulation induced the controversial results, it should be mentioned in the current study.
3. I appreciate authors well-addressed my concerns on D2 neurons. Their findings in new experiments are interesting. I suggest highlighting these findings in one new figure together with D2 neuron activity, optogenetic stimulation, and other behavioral tests or removing all of them in SI figures. (I did not see Figure R3e-f in the rebuttal letter)

Reviewer #3 (Remarks to the Author):

In general, the authors met my requests.

The experiment of c-Fos expression in NAc three weeks after sham or CCI surgery is adequate (Figure R 10). I believe neither the EMG data (Figure R9 c and d) nor the open field data (Figure R9 g and h) are golden pattern experiments to meet my concern, but they are acceptable.

The justification for not including a nociceptive test without motor bias is invalid, first because the CPP could be done with chemogenetics and second because this was just a suggestion, there are other test options. But this reviewer understands the resistance to performing experiments with totally new methodologies in an already experimentally robust article.

What this reviewer do not understand is why the authors did not include the experiments performed to follow the reviewers` requests on the MS or supplementary material. It seems that the authors simply dismissed all the queries and suggestions of the reviewers as preliminary or useless (except for Fig R1 which was included).

I'm not suggesting including all the experiments, evidently some of them are not strictly related to MS, but I would like to see at least the controls I requested and the experiments with D2 neurons (but note that Figure R 3 is not shown with fig R 6 being repeated in its place).

Point-by-point response to reviewers' comments

REVIEWER COMMENTS

Reviewer #1 (Remarks to the Author):

The authors are highly responsive to reviewers' critiques. They have performed extensive additional experiments to validate their conclusions as well as to tease apart the important differences between their findings and previously published results from various other groups. These additional results will be highly informative to the neuroscience community for guiding the proper usage of optogenetics, as well as to better understand the NAc functional neuronal ensembles. They should be all incorporated into the supplementary (or even main) figures rather than only used to show the reviewers. Related results and discussions should be also incorporated into the main text. Detailed suggestions are listed below.

Comment 1: *Response #1: This is an important clarification that needs to be incorporated into the main text. A single citation without explanation is not sufficient, as readers outside of the pain field would not immediately understand the c-Fos expression scenario here.*

Response 1: We thank the reviewer for the kind suggestion. We add “The c-Fos has been used as a neural marker of pain over several decades since Hunt et al (Hunt et al., 1987). CCI mice exhibited persistent pain hypersensitivity at least for 3 weeks. The c-Fos expression level in NAc three weeks after CCI surgery was dramatically higher than that after sham surgery (Supplementary Fig. 2a-b). Recently, the tTA-TRE (Tet-Off)-based viral system has been applied to label itch- and pain-specific neurons (Jiang et al., 2022).” in the results on Page 5, line 8-13.

Comment 2: *Response #2: “arousal” may be more suitable here than “alertness”.*

Response 2: We appreciate the reviewer for the kind suggestion. We change “alertness” for “arousal” in the discussion on Page 23, line 16.

Comment 3: *Response #6: Fig. 6H: mathematically, if POA+/total ensemble is ~ 80%, and VTA+/total ensemble is ~80%, then the overlapping subpopulation, e.g. POA+VTA+/total ensemble, should be at least 60% (80%+80%-100%=60%). The fact that it was much lower (~30%) is puzzling and needs to be addressed.*

Response 3: We apologized that we made error when performed counting. POA+NAC-ensemble+ /NAC-ensemble+ is $60.88\% \pm 1.96\%$ (mean \pm s.e.m), VTA+NAC-ensemble+ /NAC-ensemble+ is $62.52\% \pm 1.90\%$, and POA+VTA+NAC-ensemble+ /NAC-ensemble+ is $31.86\% \pm 2.52\%$. The new analysis is illustrated in new Fig. 7h, and the result is described on Page 17, line 1-2.

Comment 4: *Figure R2 demonstrates important in vivo evidence for frequency-dependent activation of NAc neurons by optogenetic stimulations. This can be incorporated into supplementary materials. It will also be helpful to specify the duration of individual laser pulses used here.*

Response 4: We thank the reviewer for the good suggestion. Figure R2 is incorporated into supplementary materials as the new Supplementary Fig. 11. The duration of individual laser pulse is 5 ms, that is described in the supplementary legends on Page 15, the last two lines. The result is described in main text on Page 13, line17-21, and Page 14, line 1-4.

Comment 5: *Figures R3 and R6 demonstrate the important differences between D2 ensemble neurons and D2 total populations in regulating pain and wakefulness. These should be included in the main and supplementary figures.*

Response 5: Figure R3 and R6 are incorporated into main text as the new Fig. 6. These results are described in main text on Page 14, line17-21 and Page 15, line1-20 and discussed on Page 22, line 8-16.

Comment 6: *Figures R4 and R5 demonstrate the important differences between terminal versus transsynaptic cell body stimulations. This should be included in the supplementary*

figures.

Response 6: Figure R4 and R5 are added into supplementary figures as the new Supplementary Fig. 13. These results are described on Page 17, line 8-21 and discussed on Page 23, line 7-10.

Comment 7: *Figure R7 complements main Figure 1 and should be included in the supplementary figures.*

Response 7: Figure R7 (NAc-ensemble-D2 Neurons are preferentially active during wakefulness) is included in the supplementary figures as the new Supplementary Fig. 12. The result is described on Page 15, line10-11.

Reviewer #2 (Remarks to the Author):

In the revised manuscript, the authors have well addressed my comments with new experiments, discussion, and illustration of previous data. I do believe that this vision has substantial improvements and seems close to be published. A few minor comments to the authors for their consideration are listed below.

Comment 1: *In Comments 1, the authors discussed the heterogeneity of NAc non-ensemble neurons to explain why activation of these non-ensemble did not alter sleep-wake behaviors. However, it is better to tune down this conclusion because activation of PVT- or VTA-projections in the NAc are also wake-promoting (these are all non-specific activation).*

Response 1: We thank the reviewer for the kind suggestion. To follow the reviewer's suggestion, we tune down the conclusion. We delete the sentence "instead of sleep and pain" in the discussion on Page 21, line 21. We add "On the other hand, the number of NAc non-ensembles neurons was relatively less compared with NAc ensemble neurons (Fig. 2b, 2h). It is also possible that activation of non-ensembles neurons (a small subset of neurons) might not affect sleep-wakefulness or pain response." in the discussion on

Page 22, line 2-5.

***Comment 2:** Comments 2, if the different frequencies of stimulation induced the controversial results, it should be mentioned in the current study.*

Response 2: We thank the reviewer for the good suggestion. Figure R2 (frequency-dependent activation of NAc D1 neurons by optogenetic stimulations) is incorporated into supplementary materials as the new Supplementary Fig. 11. The result is described in main text on Page 13, line17-21, and Page 14, line 1-4.

***Comment 3:** I appreciate authors well-addressed my concerns on D2 neurons. Their findings in new experiments are interesting. I suggest highlighting these findings in one new figure together with D2 neuron activity, optogenetic stimulation, and other behavioral tests or removing all of them in SI figures. (I did not see Figure R3e-f in the rebuttal letter)*

Response 3: We appreciate the reviewer for the kind suggestion. Figure R3 and R6 (showing the differences between D2 ensemble neurons and D2 total populations in regulating pain and wakefulness) are incorporated into main text as the new Fig. 6. These results are described on Page 14, line17-21 and Page 15, line1-20 and discussed on Page 22, line 8-16. Figure R7 (NAc-ensemble-D2 Neurons are preferentially active during wakefulness) is included in the supplementary figures as the new Supplementary Fig. 12. The result is described on Page 15, line10-11.

We apologized that we made error in Figure R3 in the last rebuttal letter. We showed Figure R6 twice but not Figure R3. The following is Figure R3. Figure R3 is incorporated into new Fig. 6.

Figure R3. Optogenetic activation of NAc-ensemble-D2 Neurons reduces both NREM sleep and pain thresholds. *a* Schematics showing the labeling and optogenetic activation of NAc-ensemble-D2 Neurons in *c-fos-tTA* transgenic mice underwent CCI surgery. *b* The strategy for specifically labelling D2 neurons in NAc-ensemble neurons by combining “Tet-Off” with “Cre-On”. This involves injecting 2 AAVs (AAV-Tre3G-CRE-WPRE-pA and rAAV-D2-DIO-hChR2-mCherry-WPREs) into NAc of *c-Fos-tTA* mice. *c* Representative EEG spectrogram (top), relative EMG trace (middle), and brain states (bottom) from a NAc-ensemble-D2-ChR2 mouse during baseline conditions and blue laser activation of NAc ensemble. Blue stripe indicates laser stimulation (473 nm, 10 Hz, 120 s). *d* Percentage of time in different brain states before, during, and after blue laser activation of NAc-ensemble-D2 Neurons (473 nm, 10 Hz, 120 s). Note the dramatical decrease in NREM sleep ($p < 0.0001$, bootstrap), increase in wakefulness ($p < 0.0001$), no obvious change in REM sleep during laser stimulation ($p > 0.05$), $n = 13$ mice. *e, f* Optogenetic activation (473 nm, 10 Hz) of NAc-ensemble-D2 Neurons significantly decreases thermal PWT ($n = 13$ mice, $F(1.605, 19.26) = 61.19$, $p < 0.0001$, BL vs. Laser, $p < 0.0001$, Pre vs. Laser, $p = 0.001$, Post vs. Laser, $p < 0.0001$) and mechanical PWT ($n = 13$ mice, $F(1.749, 20.99) = 90.74$, $p < 0.0001$, BL vs. Laser, $p < 0.0001$, Pre vs. Laser, $p = 0.0003$, Post vs. Laser, $p = 0.0036$) in *c-fos-tTA-Tg* mice. Repeated measures one-way ANOVA test with Bonferroni's multiple comparisons test for (*e, f*).

Reviewer #3 (Remarks to the Author):

In general, the authors met my requests.

Comment: *The experiment of c-Fos expression in NAc three weeks after sham or CCI surgery is adequate (Figure R 10). I believe neither the EMG data (Figure R9 c and d) nor the open field data (Figure R9 g and h) are golden pattern experiments to meet my concern, but they are acceptable.*

The justification for not including a nociceptive test without motor bias is invalid, first because the CPP could be done with chemogenetics and second because this was just a suggestion, there are other test options. But this reviewer understands the resistance to performing experiments with totally new methodologies in an already experimentally robust article.

What this reviewer do not understand is why the authors did not include the experiments performed to follow the reviewers` requests on the MS or supplementary material. It seems that the authors simply dismissed all the queries and suggestions of the reviewers as preliminary or useless (except for Fig R1 which was included).

I'm not suggesting including all the experiments, evidently some of them are not strictly related to MS, but I would like to see at least the controls I requested and the experiments with D2 neurons (but note that Figure R 3 is not shown with fig R 6 being repeated in its place).

Response: We thank the reviewer for the good suggestion. Figure R10 (The c-Fos expression in NAc three weeks after sham or CCI surgery) is added in the new Supplementary Fig. 2. This is described in the results on Page 5, line 10-12. **Please also see our response to comment 1 from reviewer #1.** Figure R9 (Optogenetic stimulations of NAc ensemble do not affect the movement of CCI-hindlimb in EMG recording and locomotion behavior in open field test.) is incorporated into supplementary materials as the new Supplementary Fig. 7. These results are described on Page 9, line 20-21 and Page 10, line 1-2. Figure R3 and R6 (showing the differences between D2 ensemble neurons and D2 total populations in regulating pain and wakefulness) are incorporated into main text as the new Fig. 6. These results are described on Page 14, line17-21 and Page 15, line1-20 and discussed on Page 22, line 8-16. Figure R7 (NAc-

ensemble-D2 Neurons are preferentially active during wakefulness) is included in the supplementary figures as the new Supplementary Fig. 12. The result is described on Page 15, line10-11. **Please also see our responses to comment 5 from reviewer #1 and comment 3 from reviewer #2.**

We apologized that we made error in Figure R3 in the last rebuttal letter. We showed Figure R6 twice but not Figure R3. The following is Figure R3. Figure R3 is incorporated into new Fig. 6.

Figure R3. Optogenetic activation of NAc-ensemble-D2 Neurons reduces both NREM sleep and pain thresholds. *a* Schematics showing the labeling and optogenetic activation of NAc-ensemble-D2 Neurons in c-fos-tTA transgenic mice underwent CCI surgery. *b* The strategy for specifically labelling D2 neurons in NAc-ensemble neurons by combining "Tet-Off" with "Cre-On". This involves injecting 2 AAVs (AAV-Tre3G-CRE-WPRE-pA and rAAV-D2-DIO-hChR2-mCherry-WPREs) into NAc of c-Fos-tTA mice. *c* Representative EEG spectrogram (top), relative EMG trace (middle), and brain states (bottom) from a NAc-ensemble-D2-ChR2 mouse during baseline conditions and blue laser activation of NAc ensemble. Blue stripe indicates laser stimulation (473 nm, 10 Hz, 120 s). *d* Percentage of time in different brain states before, during, and after blue laser activation of NAc-ensemble-D2 Neurons (473 nm, 10 Hz, 120 s). Note the dramatical decrease in NREM sleep ($p < 0.0001$, bootstrap), increase in wakefulness ($p < 0.0001$)

no obvious change in REM sleep during laser stimulation ($p > 0.05$), $n = 13$ mice. **e, f** Optogenetic activation (473 nm, 10 Hz) of NAc-ensemble-D2 Neurons significantly decreases thermal PWT ($n = 13$ mice, $F(1.605, 19.26) = 61.19$, $p < 0.0001$, BL vs. Laser, $p < 0.0001$, Pre vs. Laser, $p = 0.001$, Post vs. Laser, $p < 0.0001$) and mechanical PWT ($n = 13$ mice, $F(1.749, 20.99) = 90.74$, $p < 0.0001$, BL vs. Laser, $p < 0.0001$, Pre vs. Laser, $p = 0.0003$, Post vs. Laser, $p = 0.0036$) in *c-fos-tTA-Tg* mice. Repeated measures one-way ANOVA test with Bonferroni's multiple comparisons test for (e, f).

References

- Hunt, S.P., Pini, A., and Evan, G. (1987). Induction of c-fos-like protein in spinal cord neurons following sensory stimulation. *Nature* 328, 632-634. [10.1038/328632a0](https://doi.org/10.1038/328632a0).
- Jiang, S., Wang, Y.S., Zheng, X.X., Zhao, S.L., Wang, Y., Sun, L., Chen, P.H., Zhou, Y., Tin, C., Li, H.L., et al. (2022). Itch-specific neurons in the ventrolateral orbital cortex selectively modulate the itch processing. *Sci Adv* 8, eabn4408. [10.1126/sciadv.abn4408](https://doi.org/10.1126/sciadv.abn4408).

REVIEWERS' COMMENTS

Reviewer #1 (Remarks to the Author):

All concerns addressed. Recommend for publication.

Reviewer #3 (Remarks to the Author):

The authors met my concerns. I believe the article was significantly improved after revision and is acceptable for publication in its present form. I do not have additional requests.

Point-by-point response to reviewers' comments

REVIEWER COMMENTS

Reviewer #1 (Remarks to the Author):

Comment 1: All concerns addressed. Recommend for publication.

Response 1: We thank the reviewer for commenting our manuscript and helping us improve this study. We appreciate the reviewer's positive feedback to our revision.

Reviewer #3 (Remarks to the Author):

Comment 1: The authors met my concerns. I believe the article was significantly improved after revision and is acceptable for publication in its present form. I do not have additional requests.

Response 1: We feel thankful to the reviewer for giving us excellent advice to improve our study. We agree with the reviewer that with the new experiments in revision, our manuscript was significantly improved.